# Compression with Bayesian Implicit Neural Representations

**Zongyu Guo**\*
University of Science and
Technology of China
guozy@mail.ustc.edu.cn

**Gergely Flamich**\*
University of Cambridge
gf332@cam.ac.uk

**Jiajun He**
University of Cambridge
jh2383@cam.ac.uk

**Zhibo Chen**
University of Science and
Technology of China
chenzhibo@ustc.edu.cn

**José Miguel Hernández-Lobato**
University of Cambridge
jmh233@cam.ac.uk

## Abstract

Many common types of data can be represented as functions that map coordinates to signal values, such as pixel locations to RGB values in the case of an image. Based on this view, data can be compressed by overfitting a compact neural network to its functional representation and then encoding the network weights. However, most current solutions for this are inefficient, as quantization to low-bit precision substantially degrades the reconstruction quality. To address this issue, we propose overfitting variational Bayesian neural networks to the data and compressing an approximate posterior weight sample using relative entropy coding instead of quantizing and entropy coding it. This strategy enables direct optimization of the rate-distortion performance by minimizing the $\beta$-ELBO, and target different rate-distortion trade-offs for a given network architecture by adjusting $\beta$. Moreover, we introduce an iterative algorithm for learning prior weight distributions and employ a progressive refinement process for the variational posterior that significantly enhances performance. Experiments show that our method achieves strong performance on image and audio compression while retaining simplicity. Our code is available at https://github.com/cambridge-mlg/combiner.

## 1   Introduction

With the celebrated development of deep learning, we have seen tremendous progress of neural data compression, particularly in the field of lossy image compression [1–4]. Taking inspiration from deep generative models, especially variational autoencoders (VAEs, [5]), neural image compression models have outperformed the best manually designed image compression schemes, in terms of both objective metrics, such as PSNR and MS-SSIM [6, 7] and perceptual quality [8, 9]. However, these methods' success is largely thanks to their elaborate architectures designed for a particular data modality. Unfortunately, this makes transferring their insights *across* data modalities challenging.

A recent line of work [10–12] proposes to solve this issue by reformulating it as a model compression problem: we treat a single datum as a continuous signal that maps coordinates to values, to which we overfit a small neural network called its implicit neural representation (INR). While INRs were originally proposed in [13] to study structural relationships in the data, Dupont et al. [10] have demonstrated that we can also use them for compression by encoding their weights. Since the data

---

\*Equal Contribution.

37th Conference on Neural Information Processing Systems (NeurIPS 2023).

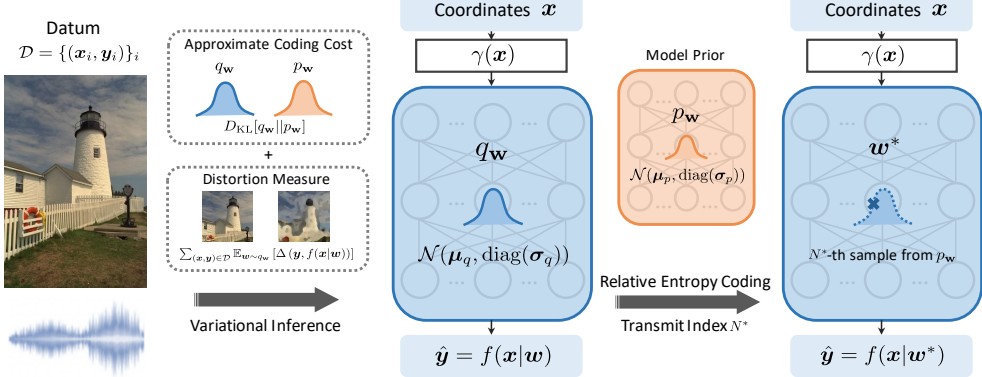

Figure 1: Framework overview of COMBINER. It first encodes a datum $\mathcal{D}$ into Bayesian implicit neural representations, as variational posterior distribution $q_{\mathbf{w}}$. Then an approximate posterior sample $\boldsymbol{w^*}$ is communicated from the sender to the receiver using relative entropy coding.

is conceptualised as an abstract signal, INRs allow us to develop universal, modality-agnostic neural compression methods. However, despite their flexibility, current INR-based compression methods exhibit a substantial performance gap compared to modality-specific neural compression models. This discrepancy exists because these methods cannot optimize the compression cost directly and simply quantize the parameters to a fixed precision, as opposed to VAE-based methods that rely on expressive entropy models [2, 3, 14–17] for end-to-end joint rate-distortion optimization.

In this paper, we propose a simple yet general method to resolve this issue by extending INRs to the variational Bayesian setting, i.e., we overfit a variational posterior distribution $q_{\mathbf{w}}$ over the weights $\mathbf{w}$ to the data, instead of a point estimate. Then, to compress the INRs, we use a relative entropy coding (REC) algorithm [18–20] to encode a weight sample $\boldsymbol{w} \sim q_{\mathbf{w}}$ from the posterior. The average coding cost of REC algorithms is approximately $D_{\mathrm{KL}}[q_{\mathbf{w}} \| p_{\mathbf{w}}]$, where $p_{\mathbf{w}}$ is the prior over the weights. Therefore, the advantage of our method is that we can directly optimize the rate-distortion trade-off of our INR by minimising its negative $\beta$-ELBO [21], in a similar fashion to VAE-based methods [22, 2]. We dub our method **Com**pression with **B**ayesian **I**mplicit **Ne**ural **R**epresentations (COMBINER), and present a high-level description of it in Figure 1.

We propose and extensively evaluate two methodological improvements critical to enhancing COMBINER's performance further. First, we find that a good prior distribution over the weights is crucial for good performance in practice. Thus, we derive an iterative algorithm to learn the optimal weight prior when our INRs' variational posteriors are Gaussian. Second, adapting a technique from Havasi et al. [23], we randomly partition our weights into small blocks and compress our INRs progressively. Concretely, we encode a weight sample from one block at a time and perform a few gradient descent steps between the encoding steps to improve the posteriors over the remaining uncompressed weights. Our ablation studies show these techniques can improve PSNR performance by more than 4dB on low-resolution image compression.

We evaluate COMBINER on the CIFAR-10 [24] and Kodak [25] image datasets and the LibriSpeech audio dataset [26], and show that it achieves strong performance despite being simpler than its competitors. In particular, COMBINER is not limited by the expensive meta-learning loop present in current state-of-the-art INR-based works [11, 12]. Thus we can directly optimize INRs on entire high-resolution images and audio files instead of splitting the data into chunks. As such, our INRs can capture dependencies across all the data, leading to significant performance gains.

To summarize, our contributions are as follows:

- We propose variational Bayesian implicit neural representations for modality-agnostic data compression by encoding INR weight samples using relative entropy coding. We call our method **Com**pression with **B**ayesian **I**mplicit **Ne**ural **R**epresentations (COMBINER).
- We propose an iterative algorithm to learn a prior distribution on the weights, and a progressive strategy to refine posteriors, both of which significantly improve performance.
- We conduct experiments on the CIFAR-10, Kodak and LibriSpeech datasets, and show that COMBINER achieves strong performance despite being simpler than related methods.

## 2 Background and Motivation

In this section, we briefly review the three core ingredients of our method: implicit neural representations (INRs; [10]) and variational Bayesian neural networks (BNNs; [21]), which serve as the basis for our model of the data, and relative entropy coding, which we use to compress our model.

**Implicit neural representations:** We can conceptualise many types of data as continuous signals, such as images, audio and video. Based on neural networks' ability to approximate any continuous function arbitrarily well [27], Stanley [13] proposed to use neural networks to represent data. In practice, this involves treating a datum $\mathcal{D}$ as a point set, where each point corresponds to a coordinate-signal value pair $(\mathbf{x}, \mathbf{y})$, and overfitting a small neural network $f(\mathbf{x} \mid \mathbf{w})$, usually a multilayer perceptron (MLP) parameterised by weights $\mathbf{w}$, which is then called the *implicit neural representation* (INR) of $\mathcal{D}$. Recently, Dupont et al. [10] popularised INRs for lossy data compression by noting that compressing the INR's weights $\mathbf{w}$ amounts to compressing $\mathcal{D}$. However, their method has a crucial shortcoming: they assume a uniform coding distribution over $\mathbf{w}$, leading to a constant *rate*, and overfit the INR only using the *distortion* as the loss. Thus, unfortunately, they can only control the compression cost by varying the number of weights since they show that quantizing the weights to low precision significantly degrades performance. In this paper, we solve this issue using variational Bayesian neural networks, which we discuss next.

**Variational Bayesian neural networks:** Based on the minimum description length principle, we can explicitly control the network weights' compression cost by making them stochastic. Concretely, we introduce a *prior* $p(\mathbf{w})$ (abbreviated as $p_{\mathbf{w}}$) and a *variational posterior* $q(\mathbf{w}|\mathcal{D})$ (abbreviated as $q_{\mathbf{w}}$) over the weights, in which case their information content is given by the *Kullback-Leibler (KL) divergence* $D_{\mathrm{KL}}[q_{\mathbf{w}} \| p_{\mathbf{w}}]$, as shown in [28]. Therefore, for distortion measure $\Delta$ and a coding budget of $C$ bits, we can optimize the constrained objective

$$\min_{q_{\mathbf{w}}} \sum_{(\boldsymbol{x}, \boldsymbol{y}) \in \mathcal{D}} \mathbb{E}_{\boldsymbol{w} \sim q_{\mathbf{w}}}[\Delta(\boldsymbol{y}, f(\boldsymbol{x} \mid \boldsymbol{w})], \quad \text{subject to } D_{\mathrm{KL}}[q_{\mathbf{w}} \| p_{\mathbf{w}}] \leqslant C. \tag{1}$$

In practice, we introduce a slack variable $\beta$ and optimize the Lagrangian dual, which yields:

$$\mathcal{L}_{\beta}(\mathcal{D}, q_{\mathbf{w}}, p_{\mathbf{w}}) = \sum_{(\boldsymbol{x}, \boldsymbol{y}) \in \mathcal{D}} \mathbb{E}_{\boldsymbol{w} \sim q_{\mathbf{w}}}[\Delta(\boldsymbol{y}, f(\boldsymbol{x} \mid \boldsymbol{w})] + \beta \cdot D_{\mathrm{KL}}[q_{\mathbf{w}} \| p_{\mathbf{w}}] + \text{const.}, \tag{2}$$

with different settings of $\beta$ corresponding to different coding budgets $C$. Thus, optimizing $\mathcal{L}_{\beta}(\mathcal{D}, q_{\mathbf{w}}, p_{\mathbf{w}})$ is equivalent to directly optimizing the rate-distortion trade-off for a given rate $C$.

**Relative entropy coding with A\* coding:** We will use *relative entropy coding* to directly encode a *single random weight sample* $\boldsymbol{w} \sim q_{\mathbf{w}}$ instead of quantizing a point estimate and entropy coding it. This idea was first proposed by Havasi et al. [23] for model compression, who introduced minimal random coding (MRC) to encode a weight sample. In our paper, we use *depth-limited, global-bound A\* coding* instead, to which we refer as A\* coding hereafter for brevity's sake [29, 20]. We present it in Appendix A for completeness. A\* coding is an importance sampling algorithm that draws[2] $N = \left\lfloor 2^{D_{\mathrm{KL}}[q_{\mathbf{w}} \| p_{\mathbf{w}}]+t} \right\rfloor$ independent samples $\boldsymbol{w}_1, \ldots, \boldsymbol{w}_N$ from the prior $p_{\mathbf{w}}$ for some parameter $t \geqslant 0$, and computes their importance weights $r_i = \log\left(q_{\mathbf{w}}(\boldsymbol{w}_i)/p_{\mathbf{w}}(\boldsymbol{w}_i)\right)$. Then, in a similar fashion to the Gumbel-max trick [31], it randomly perturbs the importance weights and selects the sample with the greatest perturbed weight. Unfortunately, this procedure returns an approximate sample with distribution $\tilde{q}_{\mathbf{w}}$. However, Theis and Yosri [32] have shown that the total variation distance $\|q_{\mathbf{w}} - \tilde{q}_{\mathbf{w}}\|_{\mathrm{TV}}$ vanishes exponentially quickly as $t \to \infty$. Thus, $t$ can be thought of as a free parameter of the algorithm that trades off compression rate for sample quality. Furthermore, A\* coding is more efficient than MRC [23] in the following sense: Let $N_{\mathrm{MRC}}$ and $N_{\mathrm{A}*}$ be the codes returned by MRC and A\* coding, respectively, when given the same target and proposal distribution as input. Then, $\mathbb{H}[N_{\mathrm{A}*}] \leqslant \mathbb{H}[N_{\mathrm{MRC}}]$, hence using A\* coding is always strictly more efficient [32].

## 3 Compression with Bayesian Implicit Neural Representations

We now introduce our method, dubbed **Com**pression with **B**ayesian **I**mplicit **Ne**ural **R**epresentations (COMBINER). It extends INRs to the variational Bayesian setting by introducing a variational posterior $q_{\mathbf{w}}$ over the network weights and fits INRs to the data $\mathcal{D}$ by minimizing Equation (2). Since

---

[2] In practice, we use quasi-random number generation with multi-dimensional Sobol sequences [30] to simulate our random variables to ensure that they cover the sample space as evenly as possible.

---

**Algorithm 1** Learning the model prior

---

**Require:** Training data $\{\mathcal{D}_i\} = \{\mathcal{D}_1, \mathcal{D}_2, ..., \mathcal{D}_M\}$.

    **Initialize** : The model posteriors $q_{\mathbf{w}}^{(i)} = \mathcal{N}(\boldsymbol{\mu}_i, \mathrm{diag}(\boldsymbol{\sigma}_i))$ of every training datum $\mathcal{D}_i$.

    **Initialize** : The model priors $p_{\mathbf{w};\boldsymbol{\theta}_p} = \mathcal{N}(\boldsymbol{\mu}_p, \mathrm{diag}(\boldsymbol{\sigma}_p))$.

    **repeat until convergence**

        **for** $i \leftarrow 1$ to $M$ **do**

            $\{q_{\mathbf{w}}^{(i)}\} \leftarrow \arg\min_{\{q_{\mathbf{w}}^{(i)}\}} \mathcal{L}(\boldsymbol{\theta}_p, \{q_{\mathbf{w}}^{(i)}\})$         ▷ Gradient descent for optimizing posteriors

        **end for**

        $\boldsymbol{\theta}_p \leftarrow \arg\min_{\boldsymbol{\theta}_p} \mathcal{L}(\boldsymbol{\theta}_p, \{q_{\mathbf{w}}^{(i)}\})$           ▷ Closed-form solution in Equation (5)

    **end repeat**

    **Return** $p_{\mathbf{w};\boldsymbol{\theta}_p} = \mathcal{N}(\boldsymbol{\mu}_p, \mathrm{diag}(\boldsymbol{\sigma}_p))$

---

encoding the model weights is equivalent to compressing the data $\mathcal{D}$, Equation (2) corresponds to jointly optimizing a given rate-distortion trade-off for the data. This is COMBINER's main advantage over other INR-based compression methods, which optimize the distortion only while keeping the rate fixed and cannot jointly optimize the rate-distortion. Moreover, another important difference is that we encode a random weight sample $w \sim q_{\mathbf{w}}$ from the weight posterior using A* coding [20] instead of quantizing the weights and entropy coding them. At a high level, COMBINER applies the model compression approach proposed by Havasi et al. [23] to encode variational Bayesian INRs, albeit with significant improvements which we discuss in Sections 3.1 and 3.2.

In this paper, we only consider networks with a diagonal Gaussian prior $p_{\mathbf{w}} = \mathcal{N}(\boldsymbol{\mu}_p, \mathrm{diag}(\boldsymbol{\sigma}_p))$ and posterior $q_{\mathbf{w}} = \mathcal{N}(\boldsymbol{\mu}_q, \mathrm{diag}(\boldsymbol{\sigma}_q))$ for mean and variance vectors $\boldsymbol{\mu}_p, \boldsymbol{\mu}_q, \boldsymbol{\sigma}_p, \boldsymbol{\sigma}_q$. Here, $\mathrm{diag}(\mathbf{v})$ denotes a diagonal matrix with $\mathbf{v}$ on the main diagonal. Following Havasi et al. [23], we optimize the variational parameters $\boldsymbol{\mu}_q$ and $\boldsymbol{\sigma}_q$ using the local reparameterization trick [33] and, in Section 3.1, we derive an iterative algorithm to learn the prior parameters $\boldsymbol{\mu}_p$ and $\boldsymbol{\sigma}_p$.

## 3.1 Learning the Model Prior on the Training Set

To guarantee that COMBINER performs well in practice, it is critical that we find a good prior $p_{\mathbf{w}}$ over the network weights, since it serves as the proposal distribution for A* coding and thus directly impacts the method's coding efficiency. To this end, in Algorithm 1 we describe an iterative algorithm to learn the prior parameters $\boldsymbol{\theta}_p = \{\boldsymbol{\mu}_p, \boldsymbol{\sigma}_p\}$ that minimize the average rate-distortion objective over some training data $\{\mathcal{D}_1, \ldots, \mathcal{D}_M\}$:

$$\bar{\mathcal{L}}_\beta(\boldsymbol{\theta}_p, \{q_{\mathbf{w}}^{(i)}\}) = \frac{1}{M} \sum_{i=1}^{M} \mathcal{L}_\beta(\mathcal{D}_i, q_{\mathbf{w}}^{(i)}, p_{\mathbf{w};\boldsymbol{\theta}_p}). \tag{3}$$

In Equation (3) we write $q_{\mathbf{w}}^{(i)} = \mathcal{N}(\boldsymbol{\mu}_q^{(i)}, \mathrm{diag}(\boldsymbol{\sigma}_q^{(i)}))$, and $p_{\mathbf{w};\boldsymbol{\theta}_p} = \mathcal{N}(\boldsymbol{\mu}_p, \mathrm{diag}(\boldsymbol{\sigma}_p))$, explicitly denoting the prior's dependence on its parameters. Now, we propose a coordinate descent algorithm to minimize the objective in Equation (3), shown in Algorithm 1. To begin, we randomly initialize the model prior and the posteriors, and alternate the following two steps to optimize $\{q_{\mathbf{w}}^{(i)}\}$ and $\boldsymbol{\theta}_p$:

1. **Optimize the variational posteriors:** We fix the prior parameters $\boldsymbol{\theta}_p$ and optimize the posteriors using the local reparameterization trick [33] with gradient descent. Note that, given $\boldsymbol{\theta}_p$, optimizing $\bar{\mathcal{L}}_\beta(\boldsymbol{\theta}_p, \{q_{\mathbf{w}}^{(i)}\})$ can be split into $M$ independent optimization problems, which we can perform in parallel:

$$\text{for each } i = 1, \ldots, M: \quad q_{\mathbf{w}}^{(i)} = \arg\min_q \mathcal{L}_\beta(\mathcal{D}_i, q, p_{\mathbf{w};\boldsymbol{\theta}_p}). \tag{4}$$

2. **Updating prior:** We fix the posteriors $\{q_{\mathbf{w}}^{(i)}\}$ and update the model prior by computing $\boldsymbol{\theta}_p = \arg\min_{\boldsymbol{\theta}} \bar{\mathcal{L}}_\beta(\boldsymbol{\theta}_p, \{q_{\mathbf{w}}^{(i)}\})$. In the Gaussian case, this admits a closed-form solution:

$$\boldsymbol{\mu}_p = \frac{1}{M} \sum_{i=1}^{M} \boldsymbol{\mu}_q^{(i)}, \quad \boldsymbol{\sigma}_p = \frac{1}{M} \sum_{i=1}^{M} [\boldsymbol{\sigma}_q^{(i)} + (\boldsymbol{\mu}_q^{(i)} - \boldsymbol{\mu}_p)^2]. \tag{5}$$

We provide the full derivation of this procedure in Appendix B. Note that by the definition of co-ordinate descent, the value of $\bar{\mathcal{L}}_\beta(\boldsymbol{\theta}_p, \{q_{\mathbf{w}}^{(i)}\})$ decreases after each iteration, which ensures that our estimate of $\boldsymbol{\theta}_p$ converges to some optimum.

## 3.2 Compression with Posterior Refinement

Once the model prior is obtained using Algorithm 1, the sender uses the prior to train the variational posterior distribution for a specific test datum, as illustrated by Equation (2). To further improve the performance of INR compression, we also adopt a progressive posterior refinement strategy, a concept originally proposed in [23] for Bayesian model compression.

To motivate this strategy, we first consider the *optimal weight posterior* $q_{\mathbf{w}}^*$. Fixing the data $\mathcal{D}$, trade-off parameter $\beta$ and weight prior $p_{\mathbf{w}}$, $q_{\mathbf{w}}^*$ is given by $q_{\mathbf{w}}^* = \arg\min_q \mathcal{L}_\beta(\mathcal{D}, q, p_{\mathbf{w}})$, where the minimization is performed over the set of all possible target distributions $q$. To compress $\mathcal{D}$ using our Bayesian INR, ideally we would like to encode a sample $\boldsymbol{w}^* \sim q_{\mathbf{w}}^*$, as it achieves optimal performance on average by definition. Unfortunately, finding $q_{\mathbf{w}}^*$ is intractable in general, hence we restrict the search over the set of all factorized Gaussian distributions in practice, which yields a rather crude approximation. However, note that for compression, we only care about encoding a **single, good quality sample** using relative entropy coding. To achieve this, Havasi et al. [23] suggest partitioning the weight vector $\mathbf{w}$ into $K$ blocks $\mathbf{w}_{1:K} = \{\mathbf{w}_1, \ldots, \mathbf{w}_K\}$. For example, we might partition the weights per MLP layer with $\mathbf{w}_i$ representing the weights on layer $i$, or into a preset number of random blocks; at the extremes, we could partition $\mathbf{w}$ per dimension, or we could just set $K = 1$ for the trivial partition. Now, to obtain a good quality posterior sample given a partition $\mathbf{w}_{1:K}$, we start with our crude posterior approximation and obtain

$$q_{\mathbf{w}} = q_{\mathbf{w}_1} \times \ldots \times q_{\mathbf{w}_K} = \underset{q_1, \ldots, q_K}{\arg\min} \mathcal{L}_\beta(\mathcal{D}, q_1 \times \ldots \times q_K, p_{\mathbf{w}}), \tag{6}$$

where each of the $K$ minimization procedures takes place over the appropriate family of factorized Gaussian distributions. Then, we draw a sample $\boldsymbol{w}_1 \sim q_{\mathbf{w}_1}$ and *refine* the remaining approximation:

$$q_{\mathbf{w}|\boldsymbol{w}_1} = q_{\mathbf{w}_2|\boldsymbol{w}_1} \times \ldots \times q_{\mathbf{w}_K|\boldsymbol{w}_1} = \underset{q_2, \ldots, q_K}{\arg\min} \mathcal{L}_\beta(\mathcal{D}, q_2 \times \ldots \times q_K, p_{\mathbf{w}} \mid \boldsymbol{w}_1), \tag{7}$$

where $\mathcal{L}_\beta(\cdot \mid \boldsymbol{w}_1)$ indicates that $\boldsymbol{w}_1$ is fixed during the optimization. We now draw $\boldsymbol{w}_2 \sim q_{\mathbf{w}_2|\boldsymbol{w}_1}$ to obtain the second chunk of our final sample. In total, we iterate the refinement procedure $K$ times, progressively conditioning on more blocks, until we obtain our final sample $\boldsymbol{w} = \boldsymbol{w}_{1:K}$. Note that already after the first step, the approximation becomes *conditionally factorized Gaussian*, which makes it far more flexible, and thus it approximates $q_{\mathbf{w}}^*$ much better [18].

**Combining the refinement procedure with compression:** Above, we assumed that after each refinement step $k$, we draw the next weight block $\boldsymbol{w}_k \sim q_{\mathbf{w}_k|\boldsymbol{w}_{1:k-1}}$. However, as suggested in [23], we can also extend the scheme to incorporate relative entropy coding, by encoding an approximate sample $\tilde{\boldsymbol{w}}_k \sim \tilde{q}_{\mathbf{w}_k|\tilde{\boldsymbol{w}}_{1:k-1}}$ with A* coding instead. This way, we actually feed two birds with one scone: the refinement process allows us to obtain a better overall approximate sample $\tilde{\boldsymbol{w}}$ by extending the variational family and by correcting for the occasional bad quality chunk $\tilde{\boldsymbol{w}}_k$ at the same time, thus making COMBINER more robust in practice.

## 3.3 COMBINER in Practice

Given a partition $\mathbf{w}_{1:K}$ of the weight vector $\mathbf{w}$, we use A* coding to encode a sample $\tilde{\boldsymbol{w}}_k$ from each block. Let $\delta_k = D_{\mathrm{KL}}[q_{\mathbf{w}_k|\tilde{\boldsymbol{w}}_{1:k-1}} \| p_{\mathbf{w}_k}]$ represent the KL divergence in block $k$ after the completion of the first $k-1$ refinement steps, where we have already simulated and encoded samples from the first $k-1$ blocks. As we discussed in Section 2, we need to simulate $\lfloor 2^{\delta_k+t} \rfloor$ samples from the prior $p_{\mathbf{w}_k}$ to ensure that the sample $\tilde{\boldsymbol{w}}_k$ encoded by A* coding has low bias. Therefore, for our method to be computationally tractable, it is important to ensure that there is no block with large divergence $\delta_k$. In fact, to guarantee that COMBINER's runtime is consistent, we would like the divergences across all blocks to be approximately equal, i.e., $\delta_i \approx \delta_j$ for $0 \leqslant i, j \leqslant K$. To this end, we set a bit-budget of $\kappa$ bits per block and below we describe the to techniques we used to ensure $\delta_k \approx \kappa$ for all $k = 1, \ldots, K$. Unless stated otherwise, we set $\kappa = 16$ bits and $t = 0$ in our experiments.

First, we describe how we partition the weight vector based on the training data, to approximately enforce our budget on average. Note that we control COMBINER's rate-distortion trade-off by

varying $\beta$ in its training loss in Equation (3). Thus, when we run Algorithm 1 to learn the prior, we also estimate the expected coding cost of the data given $\beta$ as $c_\beta = \frac{1}{M} \sum_{i=1}^{M} D_{\mathrm{KL}}[q_{\mathbf{w}}^{(i)} \| p_{\mathbf{w}}]$. Then, we set the number of blocks as $K_{\beta,\kappa} = \lceil c_\beta / \kappa \rceil$ and we partition the weight vector such that the average divergence $\bar{\delta}_k$ of each block estimated on the training data matches the coding budget, i.e., $\bar{\delta}_k \approx \kappa$ bits. Unfortunately, allocating individual weights to the blocks under this constraint is equivalent to the NP-hard bin packing problem [34]. However, we found that randomly permuting the weights and greedily assigning them using the next-fit bin packing algorithm [35] worked well in practice.

**Relative entropy coding-aware fine-tuning:** Assume we now wish to compress some data $\mathcal{D}$, and we already selected the desired rate-distortion trade-off $\beta$, ran the prior learning procedure, fixed a bit budget $\kappa$ for each block and partitioned the weight vector using the procedure from the previous paragraph. Despite our effort to set the blocks so that the average divergence $\bar{\delta}_k \approx \kappa$ in each block on the training data, if we optimized the variational posterior $q_{\mathbf{w}}$ using $\mathcal{L}_\beta(\mathcal{D}, q_{\mathbf{w}}, p_{\mathbf{w}})$, it is unlikely that the actual divergences $\delta_k$ would match $\kappa$ in each block. Therefore, we adapt the optimization procedure from [23], and we use a modified objective for each of the $k$ posterior refinement steps:

$$\mathcal{L}_{\lambda_{k:K}}(\mathcal{D}, q_{\mathbf{w} | \tilde{\mathbf{w}}_{1:k-1}}, p_{\mathbf{w}}) = \sum_{(\boldsymbol{x}, \boldsymbol{y}) \in \mathcal{D}} \mathbb{E}_{\boldsymbol{w} \sim q_{\mathbf{w}}}[\Delta(\boldsymbol{y}, f(\boldsymbol{x} \mid \boldsymbol{w})] + \sum_{i=k}^{K} \lambda_i \cdot \delta_i, \qquad (8)$$

where $\lambda_{k:K} = \{\lambda_k, \dots, \lambda_K\}$ are slack variables, which we dynamically adjust during optimization. Roughly speaking, at each optimization step, we compute each $\delta_i$ and increase its penalty term $\lambda_i$ if it exceeds the coding budget (i.e., $\delta_i > \kappa$) and decrease the penalty term otherwise. See Appendix D for the detailed algorithm.

**The comprehensive COMBINER pipeline:** We now provide a brief summary of the entire COMBINER compression pipeline. To begin, given a dataset $\{\mathcal{D}_1, \dots, \mathcal{D}_M\}$, we select an appropriate INR architecture, and run the prior learning procedure (Algorithm 1) with different settings for $\beta$ to obtain priors for a range of rate-distortion trade-offs.

To compress a new data point $\mathcal{D}$, we select a prior with the desired rate-distortion trade-off and pick a blockwise coding budget $\kappa$. Then, we partition the weight vector $\mathbf{w}$ based on $\kappa$, and finally, we run the relative entropy coding-aware fine-tuning procedure from above, using A* coding to compress the weight blocks between the refinement steps to obtain the compressed representation of $\mathcal{D}$.

## 4 Related Work

**Neural Compression:** Despite their short history, neural image compression methods' rate-distortion performance rapidly surpassed traditional image compression standards [16, 7, 9]. The current state-of-the-art methods follow a variational autoencoder (VAE) framework [2], optimizing the rate-distortion loss jointly. More recently, VAEs were also successfully applied to compressing other data modalities, such video [36] or point clouds [37]. However, mainstream methods quantize the latent variables produced by the encoder for transmission. Since the gradient of quantization is zero almost everywhere, learning the VAE encoder with standard back-propagation is not possible [38]. A popular solution [22] is to use additive uniform noise during training to approximate the quantization error, but it suffers from a train-test mismatch [39]. Relative entropy coding (REC) algorithms [19] eliminate this mismatch, as they can directly encode samples from the VAEs' latent posterior. Moreover, they bring unique advantages to compression with additional constraints, such as lossy compression with realism constraints [40, 41] and differentially private compression [42].

**Compressing with INRs:** INRs are parametric functional representations of data that offer many benefits over conventional grid-based representations, such as compactness and memory-efficiency [43–45]. Recently, compression with INRs has emerged as a new paradigm for neural compression [10], effective in compressing images [46], climate data [11], videos [47] and 3D scenes [48]. Usually, obtaining the INRs involves overfitting a neural network to a new signal, which is computationally costly [49]. Therefore, to ease the computational burden, some works [11, 46, 12] employ meta-learning loops [50] that largely reduce the fitting times during encoding. However, due to the expensive nature of the meta-learning process, these methods need to crop the data into patches to make training with second-order gradients practical. The biggest difficulty the current INR-based methods face is that quantizing the INR weights and activations can significantly degrade their performance, due to the brittle nature of the heavily overfitted parameters. Our method solves this issue

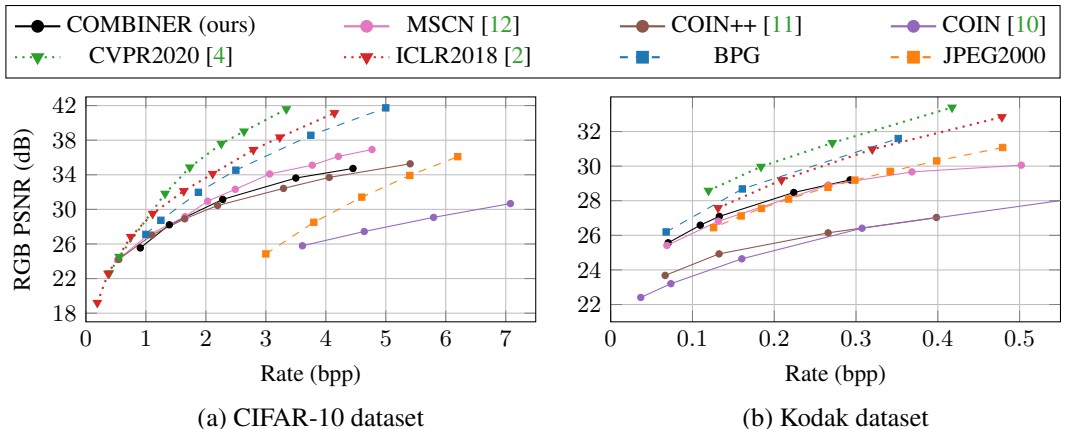

(a) CIFAR-10 dataset

(b) Kodak dataset

Figure 2: Rate-distortion curves on two image datasets. In both figures, **solid lines** denote INR-based methods, **dotted lines** denote VAE-based methods and **dashed lines** denote classical methods. Examples of decoded Kodak images are provided in Appendix F.3

by fitting a variational posterior over the parameters, from which we can encode samples directly using REC, eliminating the mismatch caused by quantization. Concurrent to our work, Schwarz et al. [51] introduced a method to learn a better coding distribution for the INR weights using a VAE, in a similar vein to our prior learning method in Algorithm 1. Their method achieves impressive performance on image and audio compression tasks, but is significantly more complex than our method: they run an expensive meta-learning procedure to learn the backbone architecture for their INRs and train a VAE to encode the INRs, making the already long training phase even longer.

## 5  Experiments

To assess COMBINER's performance across different data regimes and modalities, we conducted experiments compressing images from the low-resolution CIFAR-10 dataset [24], the high-resolution Kodak dataset [25], and compressing audio from the LibriSpeech dataset [26]; the experiments and their results are described in Sections 5.1 and 5.2. Furthermore, in Section 5.3, we present analysis and ablation studies on COMBINER's ability to adaptively activate or prune the INR parameters, the effectiveness of its posterior refinement procedure and on the time complexity of its encoding procedure.

### 5.1  Image Compression

**Datasets:** We conducted our image compression experiments on the CIFAR-10 [24] and Kodak [25] datasets. For the CIFAR-10 dataset, which contains $32 \times 32$ pixel images, we randomly selected 2048 images from the training set for learning the model prior, and evaluated our model on all 10,000 images in the test set. For the high-resolution image compression experiments we use 512 randomly cropped $768 \times 512$ pixel patches from the CLIC training set [52] to learn the model prior and tested on the Kodak images, which have matching resolution.

**Models:** Following previous methods [10–12], we utilize SIREN [43] as the network architecture. Input coordinates $x$ are transformed into Fourier embeddings [44] before being fed into the MLP network, depicted as $\gamma(x)$ in Figure 1. For the model structure, we experimentally find a 4-layer MLP with 16 hidden units per layer and 32 Fourier embeddings works well on CIFAR-10. When training on CLIC and testing on Kodak, we use models of different sizes to cover multiple rate points. We describe the model structure and other experimental settings in more detail in Appendix E. Remarkably, the networks utilized in our experiments are quite small. Our model for compressing CIFAR-10 images has only 1,123 parameters, and the larger model for compressing high-resolution Kodak images contains merely 21,563 parameters.

**Performance:** In Figure 2, we benchmark COMBINER's rate-distortion performance against classical codecs including JPEG2000 and BPG, and INR-based codecs including COIN [10], COIN++

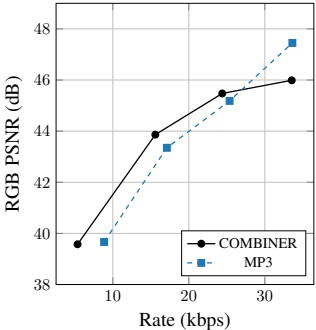
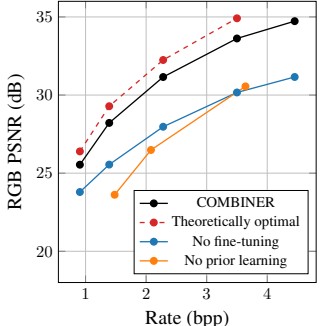
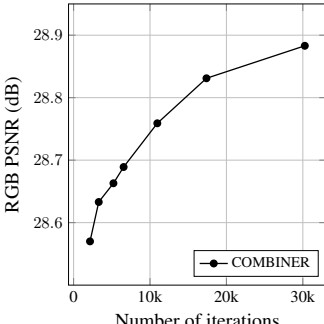

Figure 3: COMBINER's audio compression performance versus MP3 on the LibriSpeech dataset.

Figure 4: Ablation study on CIFAR-10 verifying the effectiveness of the fine-tuning and prior learning procedures.

Figure 5: COMBINER's performance improvement as a function of the number of fine-tuning steps.

[11], and MSCN [12]. Additionally, we include results from VAE-based codecs such as ICLR2018 [2] and CVPR2020 [4] for reference. We observe that COMBINER exhibits competitive performance on the CIFAR-10 dataset, on par with COIN++ and marginally lower than MSCN. Furthermore, our proposed method achieves impressive performance on the Kodak dataset, surpassing JPEG2000 and other INR-based codecs. This superior performance is in part due to our method not requiring an expensive meta-learning loop [11, 46, 12], which would involve computing second-order gradients during training. Since we avoid this cost, we can compress the whole high-resolution image using a single MLP network, thus the model can capture global patterns in the image.

## 5.2 Audio Compression

To demonstrate the effectiveness of COMBINER for compressing data in other modalities, we also conduct experiments on audio data. Since our method does not need to compute the second-order gradient during training, we can directly compress a long audio segment with a single INR model. We evaluate our method on LibriSpeech [26], a speech dataset recorded at a 16kHz sampling rate. We train the model prior with 3-second chunks of audio, with 48000 samples per chunk. The detailed experimental setup is described in Appendix E. Due to COMBINER's time-consuming encoding process, we restrict our evaluation to 24 randomly selected audio chunks from the test set. Since we lack COIN++ statistics for this subset of 24 audio chunks, we only compare our method with MP3 (implemented using the `ffmpeg` package), which has been shown to be much better than COIN++ on the complete test set [11]. Figure 3 shows that COMBINER outperforms MP3 at low bitrate points, which verifies its effectiveness in audio compression. We also conducted another group of experiments where the audios are cropped into shorter chunks, which we describe in Appendix F.2.

## 5.3 Analysis, Ablation Study and Time Complexity

**Model Visualizations:** To provide more insight into COMBINER's behavior, we visualize its parameters and information content on the second hidden layer of two small 4-layer models trained on two CIFAR-10 images with $\beta = 10^{-5}$. We use the KL in bits as an estimate of their coding cost, and do not encode the weights with A* coding or perform fine-tuning.

In Figure 6, we visualize the learned model prior parameters $\boldsymbol{\mu}_p$ and $\boldsymbol{\sigma}_p$ in the left column, the variational parameters of two distinct images in the second and third column and the KL divergence $D_{\mathrm{KL}}[q_{\mathbf{w}} \| p_{\mathbf{w}}]$ in bits in the rightmost column. Since this layer incorporates 16 hidden units, the weight matrix of parameters has a $17 \times 16$ shape, where weights and bias are concatenated (the bias is represented by the last row). Interestingly, there are seven "active" columns within $\boldsymbol{\sigma}_p$, indicating that only seven hidden units of this layer would be activated for signal representation at this rate point. For instance, when representing image 1 that is randomly selected from the CIFAR-10 test set, four columns are activated for representation. This activation is evident in the four blue columns within the KL map, which require a few bits to transmit the sample of the posterior distribution. Similarly, three hidden units are engaged in the representation of image 2. As their variational Gaussian distributions have close to zero variance, the posterior distributions at these

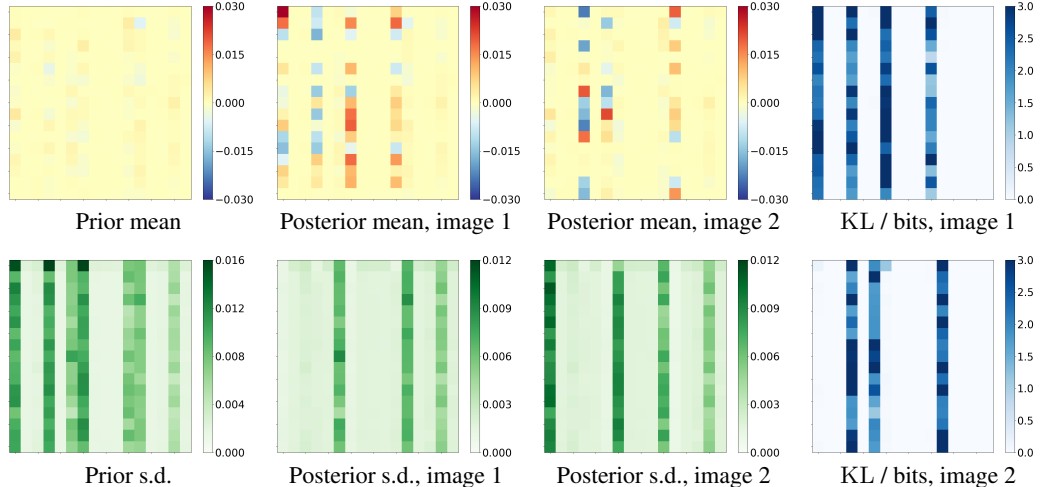

Figure 6: Visualizations of the weight prior, posterior and information content of a variational INR trained on two CIFAR-10 images. We focus on the INR's weights connecting the first and second hidden layers. Each heatmap is $17 \times 16$ because both layers have 16 hidden units and we concatenated the weights and the biases (last row). We write s.d. for standard deviation.

activated columns basically approach a Dirac delta distribution. In summary, by optimizing the rate-distortion objective, our proposed method can adaptively activate or prune network parameters.

**Ablation Studies:** We conducted ablation studies on the CIFAR-10 dataset to verify the effectiveness of learning the model prior (Section 3.1) and posterior fine-tuning (Section 3.2). In the first ablation study, instead of learning the prior parameters, we follow the methodology of Havasi ([53], p. 73) and use a layer-wise zero-mean isotropic Gaussian prior $p_\ell = \mathcal{N}(0, \sigma_\ell I)$, where $p_\ell$ is the weight prior for the $\ell$th hidden layer. We learn the $\sigma_\ell$'s jointly with the posterior parameters by optimizing Equation (3) using gradient descent, and encode them at 32-bit precision alongside the A*-coded posterior weight samples. In the second ablation study, we omit the fine-tuning steps between encoding blocks with A* coding, i.e. we never correct for bad quality approximate samples. In both experiments, we compress each block using 16 bits. Finally, as a reference, we also compare with the theoretically optimal scenario: we draw an exact sample from each blocks's variational posterior between refinement steps instead of encoding an approximate sample with A* coding, and estimate the sample's codelength with the block's KL divergence.

We compare the results of these experiments with our proposed pipeline (Section 3.3) using the above mentioned techniques in Figure 4. We find that both the prior learning and posterior refinement contribute significantly to COMBINER's performance. In particular, fine-tuning the posteriors is more effective at higher bitrates, while prior learning increases yields a consistent 4dB in gain in PSNR across all bitrates. Finally, we see that fine-tuning cannot completely compensate for the occasional bad approximate samples that A* coding yields, as there is a consistent 0.8 – 1.3dB discrepancy between COMBINER's and the theoretically optimal performance.

In Appendix C, we describe a further experiments we conducted to estimate how much each fine-tuning step contributes to the PSNR gain between compressing two blocks. The results are shown in Figure 7, which demonstrate that quality of the encoded approximate posterior sample doesn't just monotonically increase with each fine-tuning step, see Appendix C for an explanation.

**Time Complexity:** COMBINER's encoding procedure is slow, as it requires several thousand gradient descent steps to infer the parameters of the INR's weight posterior, and thousands more for the progressive fine-tuning. To get a better understanding of COMBINER's practical time complexity, we evaluate its coding time on both the CIFAR-10 and Kodak datasets at different rates and report our findings in Tables 1 and 2. We find that it can take between 13 minutes (0.91 bpp) to 34 minutes (4.45 bpp) to encode 500 CIFAR-10 images in parallel with a single A100 GPU, including posterior inference (7 minutes) and progressive fine-tuning. Note, that the fine-tuning takes longer for higher bitrates, as the weights are partitioned into more groups as each weight has higher individual information content. To compress high-resolution images from the Kodak dataset, the encoding time varies between 21.5 minutes (0.070 bpp) and 79 minutes (0.293 bpp).

| bit-rate | Encoding (500 images, GPU A100 80G) | | | Decoding (1 image, CPU) |
|---|---|---|---|---|
| | Learning Posterior | REC + Fine-tuning | Total | |
| 0.91 bpp | | ∼6 min | ∼13 min | 2.06 ms |
| 1.39 bpp | | ∼9 min | ∼16 min | 2.09 ms |
| 2.28 bpp | ∼7 min | ∼14 min 30 s | ∼21 min 30 s | 2.86 ms |
| 3.50 bpp | | ∼21 min 30 s | ∼28 min 30 s | 3.82 ms |
| 4.45 bpp | | ∼27 min | ∼34 min | 3.88 ms |

Table 1: The encoding time and decoding time of COMBINER on CIFAR-10 dataset.

| bit-rate | Encoding (1 image, GPU A100 80G) | | | Decoding (1 image, CPU) |
|---|---|---|---|---|
| | Learning Posterior | REC + Fine-tuning | Total | |
| 0.07 bpp | | ∼12 min 30 s | ∼21 min 30 s | 348.42 ms |
| 0.11 bpp | ∼9 min | ∼18 mins | ∼27 min | 381.53 ms |
| 0.13 bpp | | ∼22 min | ∼31 min | 405.38 ms |
| 0.22 bpp | | ∼50 min | ∼61 min | 597.39 ms |
| 0.29 bpp | ∼11 min | ∼68 min | ∼79 min | 602.32 ms |

Table 2: The encoding time and decoding time of COMBINER on Kodak dataset.

To assess the effect of the fine-tuning procedure's length, we randomly selected a CIFAR-10 image and encoded it using the whole COMBINER pipeline, but varied the number of fine-tuning steps between 2148 and 30260; we report the results of our experiment in Figure 5. We find that running the fine-tuning process beyond a certain point has diminishing returns. In particular, while we used around 30k iterations in our other experiments, just using 3k iterations would sacrifice a mere 0.3 dB in the reconstruction quality, while saving 90% on the original tuning time.

On the other hand, COMBINER has fast decoding speed, since once we decode the compressed weight sample, we can reconstruct the data with a single forward pass through the network at each coordinate, which can be easily parallelized. Specifically, the decoding time of a single CIFAR-10 image is between 2 ms and 4 ms using an A100 GPU, and less than 1 second for a Kodak image.

## 6 Conclusion and Limitations

In this paper, we proposed COMBINER, a new neural compression approach that first encodes data as variational Bayesian implicit neural representations and then communicates an approximate posterior weight sample using relative entropy coding. Unlike previous INR-based neural codecs, COMBINER supports joint rate-distortion optimization and thus can adaptively activate and prune the network parameters. Moreover, we introduced an iterative algorithm for learning the prior parameters on the network weights and progressively refining the variational posterior. Our ablation studies show that these methods significantly enhance the COMBINER's rate-distortion performance. Finally, COMBINER achieves strong compression performance on low and high-resolution image and audio compression, showcasing its potential across different data regimes and modalities.

COMBINER has several limitations. First, as discussed in Section 5.3, while its decoding process is fast, its encoding time is considerably longer. Optimizing the variational posterior distributions requires thousands of iterations, and progressively fine-tuning them is also time-consuming. Second, Bayesian neural networks are inherently sensitive to initialization [21]. Identifying the optimal initialization setting for achieving training stability and superior rate-distortion performance may require considerable effort. Despite these challenges, we believe COMBINER paves the way for joint rate-distortion optimization of INRs for compression.

## 7 Acknowledgements

ZG acknowledges funding from the Outstanding PhD Student Program at the University of Science and Technology of China. ZC is supported in part by National Natural Science Foundation of China under Grant U1908209, 62021001. GF acknowledges funding from DeepMind.

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

# A Relative Entropy Coding with A* Coding

---

**Algorithm 2** A* encoding

---

**Require:** Proposal distribution $p_{\mathbf{w}}$ and target distribution $q_{\mathbf{w}}$.

  **Initialize** : $N, G_0, \boldsymbol{w}^*, N^*, L \leftarrow 2^{|C|}, \infty, \perp, \perp, -\infty$

  **for** $i = 1, \ldots, N$ **do**                                 $\rhd$ $N$ samples from proposal distribution
      $\boldsymbol{w}_i \sim p_{\mathbf{w}}$
      $G_i \sim \text{TruncGumbel}(G_{i-1})$
      $L_i \leftarrow G_i + \log\left(q_{\mathbf{w}}(\boldsymbol{w}_i)/p_{\mathbf{w}}(\boldsymbol{w}_i)\right)$                $\rhd$ Perturbed importance weight
      **if** $L_i \leqslant L$ **then**
          $L \leftarrow L_i$
          $\boldsymbol{w}^*, N^* \leftarrow \boldsymbol{w}_i, i$
      **end if**
  **end for**
  **return** $\boldsymbol{w}^*, N^*$                                          $\rhd$ Transmit the index $N^*$

---

**Algorithm 3** A* decoding

---

  **Simulate** $\{\boldsymbol{w}_i\} = \{\boldsymbol{w}_1, \cdots, \boldsymbol{w}_N\}$        $\rhd$ Simulate $N$ samples from $p_{\mathbf{w}}$ with the shared seed
  **Receive** $N^*$

  **return** $\boldsymbol{w}^* \leftarrow \boldsymbol{w}_{N*}$            $\rhd$ Receive the approximate posterior sample

---

Recall that we would like to communicate a sample from the variational posterior distribution $q_{\mathbf{w}}$ using the proposal distribution $p_{\mathbf{w}}$. In our experiments, we used *global-bound depth-limited A\* coding* to achieve this [20]. We describe the encoding procedure in Algorithm 2 and the decoding procedure in Algorithm 3. For brevity, we refer to this particular variant of the algorithm as *A\* coding* for the rest of the appendix.

A* coding is an importance sampler that draws $N$ samples $\boldsymbol{w}_1, \ldots, \boldsymbol{w}_N \sim p_{\mathbf{w}}$ from the proposal distribution $p_{\mathbf{w}}$, where $N$ is a parameter we pick. Then, it computes the importance weights $r(\boldsymbol{w}_n) = q_{\mathbf{w}}(\boldsymbol{w}_n)/p_{\mathbf{w}}(\boldsymbol{w}_n)$, and sequentially perturbs them with truncated Gumbel[3] noise:

$$\tilde{r}_n = r(\boldsymbol{w}_n) + G_n, \quad G_n \sim \text{TruncGumbel}(G_{n-1}), \; G_0 = \infty \tag{9}$$

Then, it can be shown that by setting

$$N^* = \underset{n \in [1:N]}{\arg\max} \, \tilde{r}_n, \tag{10}$$

we have that $\boldsymbol{w}_{N*} \sim \tilde{q}_{\mathbf{w}}$ is approximately distributed according to the target, i.e. $\tilde{q}_{\mathbf{w}} \approx q_{\mathbf{w}}$. More preciesly, we have the following result:

**Lemma A.1** (Bound on the total variation between $\tilde{q}_{\mathbf{w}}$ and $q_{\mathbf{w}}$ (Lemma D.1 in [32]))**.** *Let us set the number of proposal samples simulated by Algorithm 2 to $N = 2^{D_{\text{KL}}[q_{\mathbf{w}} \| p_{\mathbf{w}}] + t}$ for some parameter $t \geqslant 0$. Let $\tilde{q}_{\mathbf{w}}$ denote the approximate distribution of the algorithm's output for this choice of $N$. Then,*

$$D_{TV}(q_{\mathbf{w}}, \tilde{q}_{\mathbf{w}}) \leqslant 4\epsilon, \tag{11}$$

*where*

$$\epsilon = \left( 2^{-t/4} + 2\sqrt{\mathbb{P}_{Z \sim q_{\mathbf{w}}} \left[\log_2 r(Z) \geqslant D_{\text{KL}}[Q \| P] + t/2\right]} \right)^{1/2}. \tag{12}$$

This result essentially tells us that we should draw at least around $2^{D_{\text{KL}}[q_{\mathbf{w}} \| p_{\mathbf{w}}]}$ samples to ensure low sample bias, and beyond this, the bias decreases exponentially quickly as $t \to \infty$. However,

---

[3]The PDF of a standard Gumbel random variable truncated to $(-\infty, b)$ is given by $\text{TruncGumbel}(x \mid b) = \mathbf{1}[x \leqslant b] \cdot \exp(-x - \exp(-x) + \exp(-b))$.

note that the number of samples we need also increases exponentially quickly with $t$. In practice, we observed that when $D_{\mathrm{KL}}[q_{\mathbf{w}} \| p_{\mathbf{w}}]$ is sufficiently large (around 16-20 bits), setting $t = 0$ already gave good results. To encode $N^*$, we built an empirical distribution over indices using our training datasets and used it for entropy coding to find the optimal variable-length code for the index.

In short, on the encoder side, $N$ random samples are obtained from the proposal distribution $p_{\mathbf{w}}$. Then we select the sample $\boldsymbol{w}_i$ and transmit its index $N^*$ that has the greatest perturbed importance weight. On the decoder side, those $N$ random samples can be simulated with the same seed held by the encoder. The decoder only needs to find the sample with the index $N^*$. Therefore, the decoding process of our method is very fast.

# B  Closed-Form Solution for Updating Model Prior

In this section, we derive the analytic expressions for the prior parameter updates in our iterative prior learning procedure when both the prior and the posterior are Gaussian distributions. Given a set of training data $\{\mathcal{D}_i\} = \{\mathcal{D}_1, \mathcal{D}_2, ..., \mathcal{D}_M\}$, we fit a variational distribution $q_{\mathbf{w}}^{(i)}$ to represent each of the $\mathcal{D}_i$s. To do this, we minimize the loss (abbreviated as $\mathcal{L}$ later)

$$\bar{\mathcal{L}}_{\beta}(\boldsymbol{\theta}_p, \{q_{\mathbf{w}}^{(i)}\}) = \frac{1}{M} \sum_{i=1}^{M} \mathcal{L}_{\beta}(\mathcal{D}_i, q_{\mathbf{w}}^{(i)}, p_{\mathbf{w};\boldsymbol{\theta}_p}) \tag{13}$$

$$= \frac{1}{M} \sum_{i=1}^{M} \{ \sum_{(\boldsymbol{x}, \boldsymbol{y}) \in \mathcal{D}} \mathbb{E}_{\boldsymbol{w} \sim q_{\mathbf{w}}}[\Delta(\boldsymbol{y}, f(\boldsymbol{x} \mid \boldsymbol{w})] + \beta \cdot D_{\mathrm{KL}}[q_{\mathbf{w}} \| p_{\mathbf{w};\boldsymbol{\theta}_p}] \}. \tag{14}$$

Now calculate the derivative w.r.t. the prior distribution parameter $p_{\mathbf{w};\boldsymbol{\theta}_p}$,

$$\frac{\partial \mathcal{L}}{\partial \boldsymbol{\theta}_p} = \frac{1}{M} \sum_{i=1}^{M} \frac{\partial D_{\mathrm{KL}}[q_{\mathbf{w}} \| p_{\mathbf{w}, \boldsymbol{\theta}_p}]}{\partial \boldsymbol{\theta}_p} \tag{15}$$

Considering we choose factorized Gaussian as variational distributions, the KL divergence is

$$D_{\mathrm{KL}}[q_{\mathbf{w}}^{(i)} \| p_{\mathbf{w}, \boldsymbol{\theta}_p}] = D_{\mathrm{KL}}[\mathcal{N}(\boldsymbol{\mu}_i, \mathrm{diag}(\boldsymbol{\sigma}_i)) \| \mathcal{N}(\boldsymbol{\mu}_i, \mathrm{diag}(\boldsymbol{\sigma}_i))] \tag{16}$$

$$= \frac{1}{2} \log \frac{\boldsymbol{\sigma}_p}{\boldsymbol{\sigma}_q^{(i)}} + \frac{\boldsymbol{\sigma}_q^{(i)} + (\boldsymbol{\mu}_q^{(i)} - \boldsymbol{\mu}_p)^2}{\boldsymbol{\sigma}_p} - \frac{1}{2} \tag{17}$$

To compute the analytical solution, let

$$\frac{\partial \mathcal{L}}{\partial \boldsymbol{\theta}_p} = \frac{1}{M} \sum_{i=1}^{M} \frac{\partial D_{\mathrm{KL}}[q_{\mathbf{w}} \| p_{\mathbf{w}, \boldsymbol{\theta}_p}]}{\partial \boldsymbol{\theta}_p} = 0. \tag{18}$$

Note here $\boldsymbol{\sigma}$ refers to variance rather than standard deviation. The above equation is equivalent to

$$\begin{aligned} \frac{\partial \mathcal{L}}{\partial \boldsymbol{\mu}_p} &= \sum_{i=1}^{M} \frac{\boldsymbol{\mu}_p - \boldsymbol{\mu}_q^{(i)}}{\boldsymbol{\sigma}_p} = 0, \\ \frac{\partial \mathcal{L}}{\partial \boldsymbol{\sigma}_p} &= \sum_{i=1}^{M} [\frac{1}{\boldsymbol{\sigma}_p} - \frac{\boldsymbol{\sigma}_q^{(i)} + (\boldsymbol{\mu}_q^{(i)} - \boldsymbol{\mu}_p)^2}{\boldsymbol{\sigma}_p^2}] = 0. \end{aligned} \tag{19}$$

We finally can solve these equations and get

$$\boldsymbol{\mu}_p = \frac{1}{M} \sum_{i=1}^{M} \boldsymbol{\mu}_q^{(i)}, \quad \boldsymbol{\sigma}_p = \frac{1}{M} \sum_{i=1}^{M} [\boldsymbol{\sigma}_q^{(i)} + (\boldsymbol{\mu}_q^{(i)} - \boldsymbol{\mu}_p)^2] \tag{20}$$

as the result of Equation (5) in our main text. In short, this closed-form solution provides an efficient way to update the model prior from a bunch of variational posteriors. It makes our method simple in practice, unlike some previous methods [11, 12] that require expensive meta-learning loops.

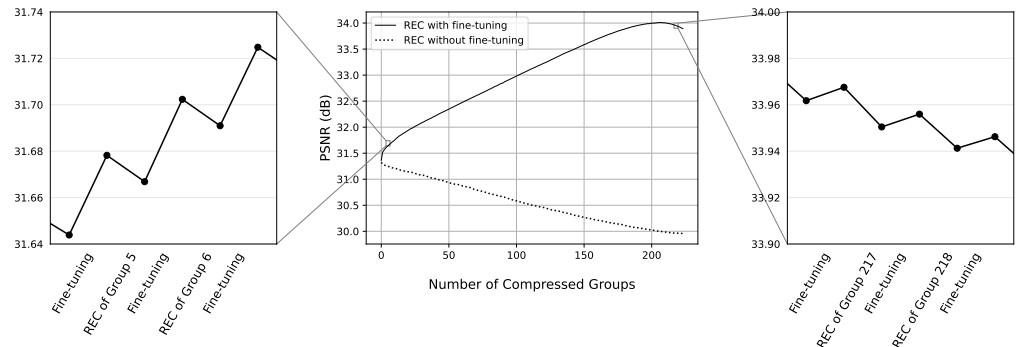

Figure 7: The approximated PSNR value changes as the fine-tuning process goes on.

## C The Approximated PSNR Changes As Fine-tuning Goes On

We compressed some of the parameters using A* coding, directly sampled the rest from the posterior distributions, and used their corresponding KL divergence to estimate the coding cost. At the same time, we can obtain the approximated PSNR value by using the posterior samples to estimate the decoding quality. As shown in Figure 7, the PSNR tends to increase as the fine-tuning process goes on. However, it tends to drop when the fine-tuning process is nearing completion. This phenomenon occurs because, at the initial fine-tuning stage, the fine-tuning gain is more than the loss from A* coding, as many uncompressed groups can be fine-tuned to correct the errors of A* coding. But when the fine-tuning process nears completion, there are fewer uncompressed groups which could compensate for the bad sample of A* coding. Therefore, the general PSNR curve tends to decrease when it approaches the end of fine-tuning. This figure shows that while A* coding's sample results may have a distance to the accurate posterior, our proposed progressive fine-tuning strategy effectively mitigates most of these discrepancies.

## D Dynamic Adjustment of $\beta$

When learning the model prior, the value of $\beta$ that controlling the rate-distortion trade-off is defined in advance to train the model prior at a specific bitrate point. After obtaining the model prior, we will first partition the network parameters into $K$ groups $\mathbf{w}_{1:K} = \{\mathbf{w}_1, \ldots, \mathbf{w}_K\}$ according to the average approximate coding cost of training data, as described in Section 3.3. Now for training the variational posterior for a given test datum, to ensure the coding cost of each group is close to $\kappa = 16$ bits, we adjust the value of $\beta$ dynamically when optimizing the posteriors. The detailed algorithm is illustrated here in Algorithm 4.

The algorithm is improved from Havasi et al. [23] to stabilize training, in the way that we set an interval $[\kappa - 0.4, \kappa]$ as buffer area where we do not change the value of $\lambda_k$. Here we only adjust $\lambda_k$ every 15 iterations to avoid frequent changes at the initial training stage.

## E Experiment Details

We introduce the experimental settings here and summarize the settings in Table 3.

### E.1 CIFAR-10

We use a 4-layer MLP with 16 hidden units and 32 Fourier embeddings for the CIFAR-10 dataset. The model prior is trained with 128 epochs to ensure convergence. Here, the term "epoch" is used to refer to optimizing the posteriors and updating the prior in the Algorithm 1 in the main text. For each epoch, the posteriors of all 2048 training data are optimized for 100 iterations using the local reparameterization trick [33], except the first epoch that contains 250 iterations. We use the Adam optimizer with learning rate 0.0002. The posterior variances are initialized as $9 \times 10^{-6}$.

---

**Algorithm 4** Dynamic $\beta$ adjustment for optimizing the posteriors

---

**Require:** $\beta, \mathbf{w}_{1:K} = \{\mathbf{w}_1, \ldots, \mathbf{w}_K\}$

  **Initialize:** $\lambda_k = \beta, k = 1, \cdots, K$
  **Initialize:** variational posterior $q_{\mathbf{w}_k}, k = 1, \cdots, K$

  **for** $i \leftarrow$ NumberIter **do**
     $\delta_k = D_{\mathrm{KL}}[q_{\mathbf{w}_k} \| p_{\mathbf{w}_k}], k = 1, \cdots, K$
     $q_{\mathbf{w}_{1:K}} \leftarrow \mathtt{VariationalUpdate}(\mathcal{L}_{\lambda_{1:K}})$     $\triangleright \mathcal{L}_{\lambda_{1:K}}$ is defined in Equation 8 in the main text
     **if** $(i \mod 15) = 0$ **then**
        **if** $\delta_k > \kappa$ **then** $\lambda_k = \lambda_k \cdot 1.05$
        **end if**
        **if** $\delta_k < \kappa - 0.4$ **then** $\lambda_k = \lambda_k \, / \, 1.05$
        **end if**
     **end if**
  **end for**
  **return** $q_{\mathbf{w}_k}, \lambda_k, k = 1, \cdots, K$

---

After obtaining the model prior, given a specific test CIFAR-10 image to be compressed, the posterior of this image is optimized for 25000 iterations, with the same optimizer. When we finally progressively compress and fine-tune the posterior, the posteriors of the uncompressed parameter groups are fine-tuned for 15 iterations with the same optimizer once a previous group is compressed.

## E.2 Kodak

For Kodak dataset, since training on high-resolution image takes much longer time, the model prior is learned using fewer training data, i.e., only 512 cropped CLIC images [52]. We also reduce the learning rate of the Adam optimizer to 0.0001 to stabilize training. In each epoch, the posterior of each image is trained for 200 iterations, except the first epoch that contains 500 iterations. We also reduce the total epoch number to 96 which is empirically enough to learn the model prior.

We use two models with different capacity for compressing high-resolution Kodak images. The smaller model is a 6-layer SIREN with 48 hidden units and 64 Fourier embeddings. This model is used to get the three low-bitrate points in Figure 2b in our main text, where the corresponding beta is set as $\{10^{-7}, 10^{-8}, 4 \times 10^{-8}\}$. Another larger model comprises a 7-layer MLP with 56 hidden units and 96 Fourier embeddings, which is used for evaluation at the two relatively higher bitrate points in Figure 2b in our main text. The betas of these two models have the same value $2 \times 10^{-9}$. We empirically adjust the variance initialization from the set $\{4 \times 10^{-6}, 4 \times 10^{-10}\}$ and find they can affect the converged bitrate and achieve good performance. In particular, the posterior variance is initialized as $4 \times 10^{-10}$ to reach the highest bitrate point in the rate-distortion curve. The posterior variance of other bitrate-points on Kodak dataset are all initialized as $4 \times 10^{-6}$.

**Important note:** It required significant empirical effort to find the optimal parameter settings we described above, hence our note in the Conclusion and Limitations section that Bayesian neural networks are inherently sensitive to initialization [21].

## E.3 LibriSpeech

We randomly crop 1024 audio samples from LibriSpeech "train-clean-100" set [26] for learning the model prior and randomly crop 24 test samples from "test-clean" set for evaluation. The model structure is the same as the small model used for compressing Kodak images. We evaluate on four bitrate points by setting $\beta = \{10^{-7}, 3 \times 10^{-8}, 10^{-8}, 10^{-9}\}$. There are 128 epochs, and each epoch has 100 iterations with learning rate as 0.0002. The first epoch has 250 iterations. In addition, the posterior variance is initialized as $4 \times 10^{-9}$. The settings for optimizing and fine-tuning posterior of a test datum are the same as the experiments on Kodak dataset.

| | CIFAR-10 | Kodak | | LibriSpeech |
|---|---|---|---|---|
| | | Smaller Model | Larger Model | |
| Network Structure | | | | |
| number of MLP layer | 4 | 6 | 7 | 6 |
| hidden unit | 16 | 48 | 56 | 48 |
| Fourier embedding | 32 | 64 | 96 | 64 |
| number of parameters | 1123 | 12675 | 21563 | 12675 |
| Learning Model Prior from Training Data | | | | |
| number of training data | 2048 | 512 | 512 | 1024 |
| epoch number | 128 | 96 | 96 | 128 |
| learning rate | 0.0002 | 0.0001 | 0.0001 | 0.0002 |
| iteration / epoch (except the first epoch) | 100 | 200 | 200 | 100 |
| iteration number in the first epoch | 250 | 500 | 500 | 250 |
| initialization of posterior variance | $9 \times 10^{-6}$ | $4 \times 10^{-6}$ | $4 \times 10^{-6}, 4 \times 10^{-10}$ | $4 \times 10^{-9}$ |
| $\beta$ | $2 \times 10^{-5}, 5 \times 10^{-6}, 2 \times 10^{-6}$ $1 \times 10^{-6}, 5 \times 10^{-7}$ | $10^{-7}, 10^{-8}, 4 \times 10^{-8}$ | $4 \times 10^{-6}$ | $10^{-7}, 3 \times 10^{-8}$ $10^{-8}, 10^{-9}$ |
| Optimize the Posterior of a Test Datum | | | | |
| iteration number | 25000 | 25000 | 25000 | 25000 |
| learning rate | 0.0002 | 0.0001 | 0.0001 | 0.0002 |
| training with 1/4 the points (pixels) | ✗ | ✓ | ✓ | ✗ |
| number of group (KL budget = 16 bits / group) | (58, 89, 146, 224, 285) | (1729, 2962, 3264) | (5503, 7176) | (1005, 2924, 4575, 6289) |
| bitrate, (bpp for images, Kbps for audios) | (0.91, 1.39, 2.28, 3.50, 4.45) | (0.070, 0.110, 0.132) | (0.224, 0.293) | (5.36, 15.59, 24.40, 33.54) |
| PSNR, dB | (0.91, 1.39, 2.28, 3.50, 4.45) | (0.070, 0.110, 0.132) | (0.224, 0.293) | (5.36, 15.59, 24.40, 33.54) |

Table 3: Hyper parameters in our experiments.

# F  Supplementary Experimental Results

## F.1  Number of Training Samples

Since the model prior is learned from a few training data, the number of training data may influence the quality of the learned model prior. We train the model prior with a different number of training images from the CIFAR-10 training set and evaluate the performance on 100 randomly selected test images from the CIFAR-10 test set. Surprisingly, as shown in Figure 8, we found that even merely 16 training images can help to learn a good model prior. Considering the randomness of training and testing, the performance on this test subset is almost the same when the number of training data exceeds 16. This demonstrates that the model prior is quite robust and generalizes well to test data. In our final experiments, the number of training samples is set to 2048 (on CIFAR-10 dataset) to ensure the prior converges to a good optimum.

## F.2  Compressing Audios with Small Chunks

The proposed approach does not need to compute the second-order gradient during training, which helps to learn the model prior of the entire datum. Hence, compression with a single Bayesian INR network helps to fully capture the global dependencies of a datum. That is the reason for our strong performance on Kodak and LibriSpeech datasets. Here, we also conduct a group of experiment to compare the influence of cropping audios into chunks. Unlike the experimental setting in our main text that compresses every 3-second audio ($1 \times 48000$) with a single MLP network, here we try to crop all the 24 audios into small chunks, each of the chunk has the shape of $1 \times 200$. We use the same

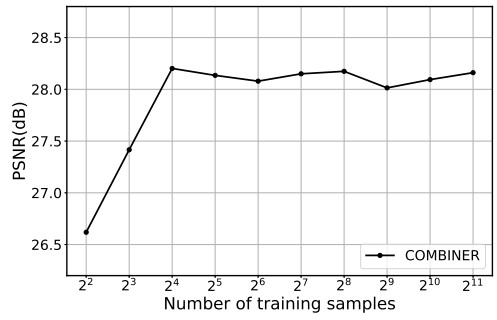 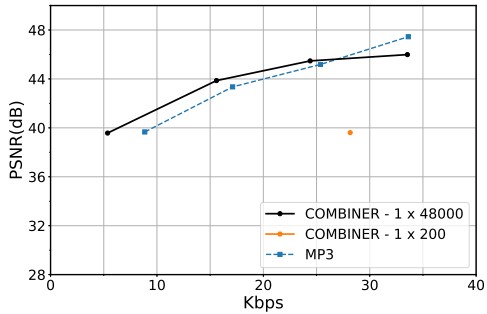

Figure 8: Impact of the number of training data.  Figure 9: Compressing audios.

network used for compressing CIFAR-10 images for our experiments here. As shown in Figure 9, if we do not compress the audio as an entire entity, the performance will drops for around 5 dB. It demonstrates the importance of compressing with a single MLP network to capture the inherent redundancies within the entire data.

### F.3 Additional Figures

We provide some examples of the decoded Kodak images in Figure 10.

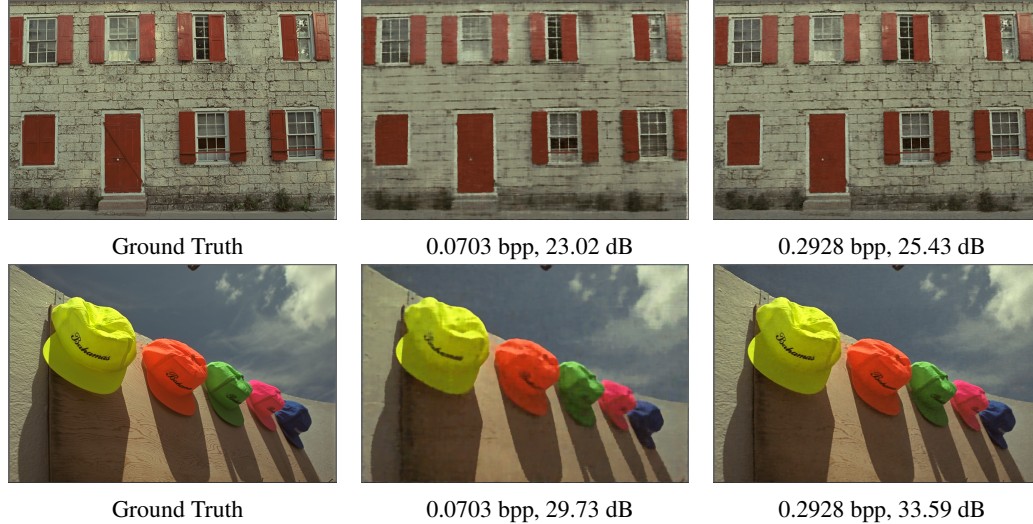

| Ground Truth | 0.0703 bpp, 23.02 dB | 0.2928 bpp, 25.43 dB |

| Ground Truth | 0.0703 bpp, 29.73 dB | 0.2928 bpp, 33.59 dB |

Figure 10: Decoded Kodak images.

