# Appendix - Compression with Bayesian Implicit Neural Representations

1 In addition to the four appendix sections mentioned in our main paper, we would like to draw atten-
2 tion to two additional experiments: one evaluating the practical training and coding time, and the
3 other investigating the impact of the number of training samples. These two experiments, especially
4 the later one, offer **crucial** insights and are detailed in Appendix E1 and Appendix E2, respectively.

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

## E    Supplementary Experimental Results

### E.1    Evaluation of Training and Practical Coding Time

**Training Time.** The most time-consuming part of setting up COMBINER for a data modality is running the iterative prior learning algorithm we propose in the main text. Furthermore, optimizing the variational posteriors takes up the bulk of the learning process since updating the prior parameters given the variational posteriors can be done efficiently using the formulae we derive in Appendix B. However, such posterior optimization can be done in parallel, especially given that our INRs have very few parameters. To train the model prior on the CIFAR-10 dataset, we can train the posteriors of 2048 images together in a single V100 GPU. It only takes around **20 minutes** to train for 128 epochs with almost 100 iterations per epoch. For training the model prior with 512 cropped CLIC images, due to the limit of GPU memory, we run multiple processes simultaneously on 4 GTX 1080 GPUs, where each process runs for a single image. The entire training time on CLIC dataset consumes around 30 hours. We note that the training time on CLIC dataset could be significantly reduced with additional engineering effort, but we have not had the opportunity to do so due to time constraints.

**Coding Time.** To compress a test datum, we first optimize its INR's variational posterior for 25,000 iterations. Such optimization process should also be included as a part of encoding time, similar to COIN [10]. In addition, the progressive posterior refinement process also takes a long time. Therefore, the encoding time of our method is very long. Note that the encoding time of relative entropy coding is negligible compared with the optimization process because our model is very small, and there are not so many parameter groups. As a result, we are able to evaluate all the 10,000 images from the CIFAR-10 test set in parallel using a CPU cluster. To decode the network parameters, the decoder only needs to search for the sample according to the received index, which is very fast. The practical encoding and decoding time is shown in Table 2.

|  | CIFAR, bpp = 0.91 | CIFAR, bpp = 4.45 | Kodak, bpp = 0.070 | Kodak, bpp = 0.293 |
|---|---|---|---|---|
| encoding time | ∼10 minutes | ∼20 minutes | ∼2 hours | ∼4 hours |
| decoding time | 0.051 second | 0.075 second | 0.410 second | 0.542 second |

Table 2: Practical encoding and decoding time of a specific image on 1080Ti GPU.

We show both the encoding and decoding time of our method on different datasets at different bitrates. In fact, the decoding time is mainly consumed for relative entropy decoding and inference

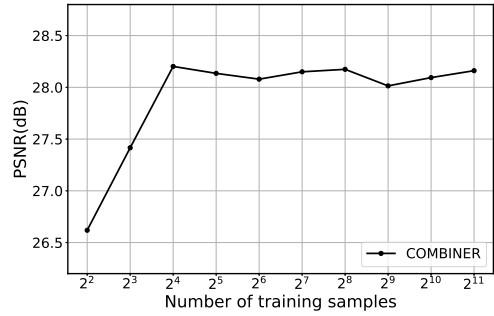

Figure 1: Impact of the number of training data.

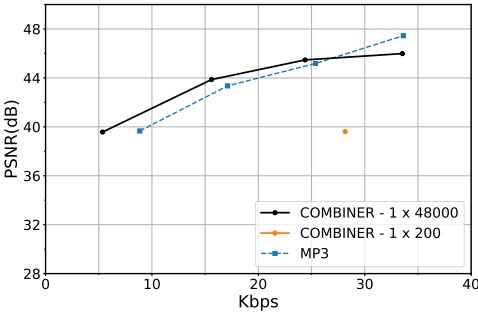

Figure 2: Compressing audios.

of the received MLP network. Usually, if there are more parameter groups, the coding time will be longer. Therefore, decoding a Kodak image at our highest bitrate (0.293 bpp) consumes the most decoding time (0.542 second), but is still very fast.

## E.2 Number of Training Samples

Since the model prior is learned from a few training data, the number of training data may influence the quality of the learned model prior. We train the model prior with a different number of training images from the CIFAR-10 training set and evaluate the performance on 100 randomly selected test images from the CIFAR-10 test set. Surprisingly, as shown in Figure 1, we found that even merely 16 training images can help to learn a good model prior. Considering the randomness of training and testing, the performance on this test subset is almost the same when the number of training data exceeds 16. This demonstrates that the model prior is quite robust and generalizes well to test data. In our final experiments, the number of training samples is set to 2048 (on CIFAR-10 dataset) to ensure the prior converges to a good optimum.

## E.3 Compressing Audios with Small Chunks

The proposed approach does not need to compute the second-order gradient during training, which helps to learn the model prior of the entire datum. Hence, compression with a single Bayesian INR network helps to fully capture the global dependencies of a datum. That is the reason for our strong performance on Kodak and LibriSpeech datasets. Here, we also conduct a group of experiment to compare the influence of cropping audios into chunks. Unlike the experimental setting in our main text that compresses every 3-second audio ($1 \times 48000$) with a single MLP network, here we try to crop all the 24 audios into small chunks, each of the chunk has the shape of $1 \times 200$. We use the same network used for compressing CIFAR-10 images for our experiments here. As shown in Figure 2, if we do not compress the audio as an entire entity, the performance will drops for around 5 dB. It demonstrates the importance of compressing with a single MLP network to capture the inherent redundancies within the entire data.

## E.4 Additional Figures

We provide some examples of the decoded Kodak images in Figure 3.

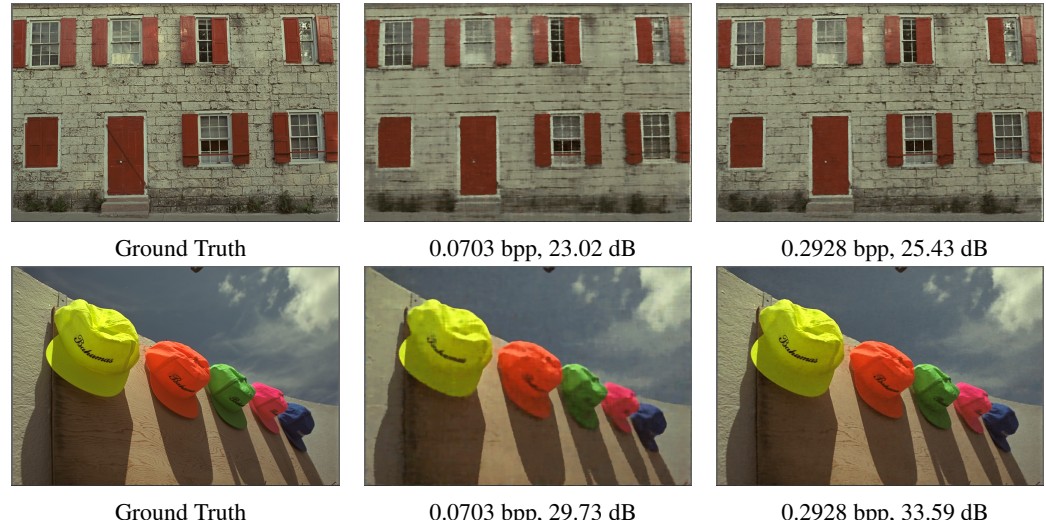

| Ground Truth | 0.0703 bpp, 23.02 dB | 0.2928 bpp, 25.43 dB |
| Ground Truth | 0.0703 bpp, 29.73 dB | 0.2928 bpp, 33.59 dB |

Figure 3: Decoded Kodak images.