# OpenReview forum: "Compression with Bayesian Implicit Neural Representations"
_NeurIPS.cc/2023/Conference — NeurIPS 2023 spotlight_

### Official Review · Reviewer_rYFn · 2023-06-24

**Soundness:** 2 fair
**Presentation:** 3 good
**Contribution:** 3 good
**Rating:** 7
**Confidence:** 4

**Summary:**

## Summary
The authors propose an approximated correlation communication approach to compress Bayesian INF. Two practical considerations, such as prior fitting and posterior refinement are proposed. It achieves an R-D performance comparable to the SOTA INF method on image compression. Moreover, the audio compression is also considered.

========================================
## Post rebuttal
The authors have addressed my concern wrt to the experiments, in a quite unexpected way. I raise my rating to accept.


**Strengths:**

## Strength
* The idea of using correlation communication to compress image INR is novel. The compression of bayesian INR is also novel and likely to have impact outside compression community. Furthermore, it also considers audio compression and achieves reasonable results.


**Weaknesses:**

## Weakness
* To the best of my knowledge, there is no algorithm that communicates correlation using finite number of samples. The adaptative reject sampling [Harsha 2010, The Communication Complexity of Correlation] and Possion Functional Representation [Li 2018, Strong Functional Representation Lemma and Applications to Coding Theorems] both requires infinite number of samples in the worst case to communicate exact posterior. As [Theis 2021, Algorithms for the Communication of Samples] has shown in Fig. 2, the expected code length of relative entropy coding [Flamich 2020], even if converge expotentially fast, still have quite some distance to the true posterior. And the expected code length of [Flamich 2020] is also inferior to [Harsha 2010] [Li 2018], which is $D + \log D + \log\log D +,... $ The [Flamich 2020] does not emphasis this gap, while it is supposed to negatively effect the performance of the proposed approach. And this inaccurate posterior gap also exists for A* coding [Flamich 2022]. Then my question is, how bad such posterior approximation it is? Or to say, if we assume that we can use infinite sample optimal PFR to achieve exact posterior, what is the R-D curve like?
* Another weakness is that the advantage of the proposed approach over MSCN [Schwarz 2022, Meta-Learning Sparse Compression Networks] is unclear. From the perspective of R-D performance, the proposed approach is only comparable to MSCN on Kodak and is slightly outperformed by MSCN on CIFAR. However, the encoding time of the proposed approach seems to be quite long due to the progressive posterior refinement. As neither the author nor original MSCN paper report encoding time of MSCN, I can not evaluate the advantage of the proposed approach pver MSCN. Or to say, if the authors can show that the run time of propoased approach is significantly faster than MSCN, this paper is more acceptable. This is especially true as the authors claim that their approach is simpler (L54,66,253). They need evdience, such as faster runtime, to support their claim.

**Questions:**

See weakness.

**Limitations:**

The limtation is discussed.

---

> ### Author Rebuttal · Authors · 2023-08-09
>
> We thank the reviewer for the thoughtful review and the valuable questions. We respond to their concerns below.
>
> **Concerns regarding channel simulation:** While the reviewer states important questions about the quality of our coding scheme's posterior approximation quality (which we address below), we also note that the reviewer conflates exact and approximate channel simulation. Harsha et al.'s rejection sampler, PFR and A* coding are all exact channel simulation algorithms, whose optimal, practically achievable expected codelengths are at most $\mathbb{I}[\mathcal{D} : w] + \log(\mathbb{I}[\mathcal{D} : w] + 1) + 5$ bits (Theorem 1; Li & El Gamal, 2018), where $\mathbb{I}[\mathcal{D} : w]$ denotes the mutual information between the data $\mathcal{D}$ and the INR weights $w$. Furthermore, given mild constraints that are virtually always satisfied in practice (bounded target-proposal density ratio), all these algorithms terminate with probability 1. However, as we mention in line 96, we use depth-limited A* coding in our work, which is an approximate channel simulation algorithm with finite runtime and the bias of which is formally characterised by Theorem 3.2 of Havasi et al. [23].
> > How bad such posterior approximation it is? Or to say, if we assume that we can use infinite sample optimal PFR to achieve exact posterior, what is the R-D curve like?
>
> We thank the reviewer for this question. First, as we explain in lines 174-180, we use our finetuning procedure precisely to mitigate the approximation gap. However, we have also added an extra group of experiments to show the theoretical R-D curve in Figure 2 in the rebuttal document that we will also include in the camera-ready version of our paper. In these new experiments, we still finetune the model parameters progressively. However, instead of A* coding, we directly sample from the posterior distribution of the current block and assume that the sample is transmitted to the decoder at maximum efficiency, using $D_{KL}[Q || P]$ bits. We see that on the CIFAR-10 dataset, the theoretically optimal performance is 1 dB higher than the practical performance, demonstrating that there is still much room for improvement for our coding scheme.
>
> In addition, we conducted another experiment to investigate how the PSNR value changes during the finetuning process, the results of which are shown in the rebuttal document as Figure 3. We compressed some of the parameters using A* coding, directly sampled the rest from the posterior distributions, and used their corresponding KL divergence to estimate the coding cost. Interestingly, the PSNR tends to increase as the finetuning process goes on, however, it tends to drop when the finetuning process is nearing completion. This phenomenon occurs because, at the initial finetuning stage, the finetuning gain is more than the loss from A* coding, as many uncompressed groups can be finetuned to correct the errors of A* coding. But when the finetuning process nears completion, there are fewer uncompressed groups which could compensate for the bad sample of A* coding. Therefore, the general PSNR curve tends to decrease when it approaches the end of finetuning.
>
> In summary, these two figures show that while A* coding's sample results may have a distance to the accurate posterior, our proposed progressive finetuning strategy effectively mitigates most of these discrepancies.
> > Another weakness is that the advantage of the proposed approach over MSCN is unclear.
>
> We recently caught an error in the evaluation of the MSCN paper, which we confirmed with the authors in private communication. In particular, the image compression performance they report on the Kodak dataset has some errors. Since they cannot directly apply their method to the high-resolution images due to computational constraints, they split the images into patches and encoded them separately. However, they calculate the PSNR for the images by averaging the PSNR values of their patches, which is incorrect. The correct would have been to reassemble the patches first and compute the PSNR on the whole image. Thus, COMBINER actually performs much better than MSCN on the Kodak dataset (more than 1.5dB gain).
> > As the authors claim that their approach is simpler, they need evidence, such as faster runtime, to support their claim.
>
> We thank the reviewer for their suggestion, we elaborate on COMBINER's simplicity and runtime below.
>
> **Simplicity:** Our claim refers to the technical simplicity of our method, which consists only of two parts: (1) train a variational Bayesian implicit neural representation on some data and (2) transmit a sample from the INR's weight posterior to encode the data it represents.
>
> **Runtime:** After improving the practical implementation of entropy coding, we update our method's encoding and decoding times in Tables 1 and 2, as shown in the one-page rebuttal document. Now both the encoding time and decoding time are acceptable for practical application. Specifically, on the CIFAR-10 dataset, it only takes around 33 minutes for our largest bitrate model to encode 500 images from the CIFAR-10 dataset, meaning each CIFAR-10 image takes around 3.96 seconds for encoding. We can compare our method COMBINER with previous methods like COIN [10] and COIN++ [11] (according to the coding time reported in the paper of COIN++), showing the comparisons in the below table.
>
> |                   |  COIN             |  COIN++           |  COMBINER (Ours)  |
> |-------------------|-------------------|-------------------|-------------------|
> | Encoding Time     |    2.97 s         |    0.095 s        |    3.96 s         |
> | Decoding Time     |    0.46 ms        |    1.29 ms        |    3.90 ms        |
>
> _We are happy to address any further questions the reviewer might have. However, should we have addressed the reviewer's concerns adequately, we kindly ask them to consider raising their score._

---

> > ### Comment · Reviewer_rYFn · 2023-08-14
> >
> > * First of all, if exact posterior is used, the the rate shuold be $D_{KL} + \log D_{KL} + \log\log D_{KL}...$ bits. If the current curve of Figure 2 in the rebuttal document, it is wrong, as its rate is $D_{KL}$ bits, which is only true for asymptomatic channel simulation, not one shot channel simulation like this paper. I suggest the author to consider this one-shot overhead and show the correct figure in the later version.
> > * Second, I am glad to see that the authors address my concern related to MSCN, in a very very unexpected way. Currently I am more confident with accepting this paper, and I will raise my rating to accept.

---

> > > ### Author Response · Authors · 2023-08-14
> > > **Response to Reviewer rYFn**
> > >
> > > We thank the reviewer for their response to our rebuttal; we address the reviewers first comment below
> > >
> > > > First of all, if exact posterior is used, the the rate shuold be $D_{KL} + \log D_{KL} + \log\log D_{KL} + \dots$ bits. If the current curve of Figure 2 in the rebuttal document, it is wrong, as its rate is $D_{KL}$ bits, which is only true for asymptomatic channel simulation, not one shot channel simulation like this paper. I suggest the author to consider this one-shot overhead and show the correct figure in the later version.
> > >
> > > We believe there are two points of slight misunderstanding here, which we address below. Please let us know if we did not interpret the reviewer's comment correctly.
> > >
> > > **The reported curve in Figure 2:** We believe the reviewer assumes that the dashed line in Figure 2 is supposed to describe the efficiency of an exact channel simulation protocol, for which the codelength would include some additional overhead terms besides the KL divergence, as the reviewer rightly mentions. However, this is not our intention. As described in Section 3.3 of the paper, we use an approximate protocol such that the codelength is always approximately equal to the KL (within each block). Hence, what we compare in Figure 2 is the quality of approximate samples compared to exact ones at the same coding cost. In other words, the dashed line describes the practically unattainable scenario if our scheme would always yield exact samples instead of approximate ones at the same codelength as before.
> > >
> > > We also note that if we included the extra log term in the rate calculation of the idealised sampler, the theoretically ideal curve would look worse since it would shift the curve to the right; thereby making COMBINER look better.
> > >
> > > **The overhead terms in exact channel simulation:** We believe the reviewer is referring to the limiting case of Li and Vitanyi's universal prefix-free codes to encode the index returned by the sampler, yielding a codelength bound of approximately $D_{KL} + \log D_{KL} + \log\log D_{KL} + \dots$ in the one-shot case. Via Jensen's inequality, this would yield a bound of approximately $I[\mathcal{D} : w] + \log I[\mathcal{D} : w]  + \log\log I[\mathcal{D} : w] + \dots$ in the average case, where $I[\mathcal{D} : w]$ denotes the mutual information between the data $\mathcal{D}$ and the INR weights $w$ that encode it. However, using the zeta distribution approach described in Appendices A and B in Li & El Gamal (2018), we can reduce the average-case bound to approximately $I[\mathcal{D} : w] + \log (I[\mathcal{D} : w] + 1) + 5$, which is a better bound!
> > >
> > > ## References
> > >
> > > M. Li and P. Vitanyi, An Introduction to Kolmogorov Complexity and
> > > Its Applications, 3rd ed. New York: Springer-Verlag, 2008.
> > >
> > > C. T. Li and A. El Gamal (2018). Strong functional representation lemma and applications to coding theorems. IEEE Transactions on Information Theory, 64(11), 6967-6978.

---

### Official Review · Reviewer_5R2p · 2023-06-27

**Soundness:** 3 good
**Presentation:** 2 fair
**Contribution:** 3 good
**Rating:** 6
**Confidence:** 4

**Summary:**

This paper proposes overfitting variational Bayesian neural networks to the data and compressing an approximate posterior weight sample using relative entropy coding, which enables direct optimization of the rate-distortion performance by minimizing the $\beta$-ELBO. Moreover, an iterative algorithm for learning prior weight distributions is introduced and a progressive refinement process for the variational posterior is employed for improved performance. Experiments were conducted on image and audio compression.

**Strengths:**

- the work encodes data with variational Bayesian implicit neural representations, which enables direct optimization of the rate-distortion performance by minimizing the $\beta$-ELBO.
- the work presents an iterative algorithm for learning prior weight distributions and a progressive refinement process for the variational posterior for improved performance.


**Weaknesses:**

The presentation can be improved, e.g.,
- $\theta_p$ indicates $\mu_p, \sigma_p$, which should be clearly described.
- Some typos, e.g., Line 183: represent -> represents; Line 190: the to -> the; Line 209: the its -> its; Line 296: subscript "2".
- Title of [5] is missed, "Auto-Encoding Variational Bayes". Similar problem in [35]. And [35] actually repeats [34].

**Questions:**

- It would be nice to compare the proposed method with the concurrent work [52[ published in ICML 2023.
- Will the encoding time be the obstacle for applying he porposed method to video compression or 3D data compression?
- Code availability helps reproducibility although enough implementation details are provided.


**Limitations:**

The authors adequately addressed the limitations.

---

> ### Author Rebuttal · Authors · 2023-08-09
>
> We thank the reviewer for the careful reading and detailed review. We will carefully proofread the paper again and correct the typos and references in our final version. Moreover, we address the reviewer's questions below.
>
> > It would be nice to compare the proposed method with the concurrent work [52] published in ICML 2023.
>
> VC-INR [52] adopts a similar MAML framework also used by prior works, including MSCN [13] and COIN++ [11], and achieves impressive results. Our approach is novel and different from all these works; thus, we still have a gap with that paper. However, we would like to note the significant gap in complexity between the two methods: besides using the MAML-based approach, VC-INR uses a complex gating mechanism, low-rank weight factorization and variational auto-encoders to encode the weight factors to obtain its good performance. On the other hand, COMBINER simply encodes a variational INR with relative entropy coding.
>
> > Will the encoding time be the obstacle for applying the proposed method to video compression or 3D data compression?
>
> Thanks for your question. Although videos or 3D data usually have more pixels/points than image data, our method still applies to these data formats. As a simple, practical solution, we can sample a subset of pixels during training without significantly compromising performance. For example, when training the model prior on a high-resolution image dataset, we sample 25% pixels to accelerate training, which we found to have negligible impact on the final performance. On the other hand, we can reduce the iteration number during the progressive finetuning process. As shown in Figure 1 in the rebuttal document, as we decrease the iteration number, the finetuning time reduces from nearly 1 hour (corresponding to nearly 30000 iterations) to 334 seconds (less than 3000 iterations), which only sacrifices 0.3dB reconstruction quality.
>
> > Code availability helps reproducibility although enough implementation details are provided.
>
> It took us some time to clean up the code after submitting the manuscript. We have completed cleaning up part of the code and sent the anonymous link to the AC guided by the NeurIPS rebuttal policy.

---

> > ### Comment · Reviewer_5R2p · 2023-08-14
> > **Thank you for the response**
> >
> > Thank you for the response.

---

### Official Review · Reviewer_7dCG · 2023-07-07

**Soundness:** 3 good
**Presentation:** 3 good
**Contribution:** 3 good
**Rating:** 6
**Confidence:** 4

**Summary:**

The paper improves INR-based (image) compression by introducing majorly two techniques: 1) a relative entropy coding based model compressing framework as an alternative to commonly used quantization - entropy coding pipeline; 2) a semi-amortized approach to train the model prior which is similar to beta-VAE and enables RD tradeoff with relative entropy coding. With the proposed novel pipeline, one can adopt COIN-like INR compression with better RD performance because it gets rid of quantization errors.

**Strengths:**

The paper is well-written with dedicated figures and easy to follow. Overall I like the story of adopting beta-VAE/REC in INR coding methods. It is intuitive that given a dataset, all data neural representations share some common knowledge, and a potential amortized approach tends to exist. Thus, a meta-model describing the prior model distribution is soundable. This re-connects compression with VAEs, though a different way from feature-based learned image compression methods. And this re-connection makes a finer rate control of INR more promising. I believe this work contributes to society and can inspire future studies.

**Weaknesses:**

My major concern is about practicality. Seems that to train a model prior, we should first train many model posteriors. As also discussed by the authors, this training makes the entire training (encoding) time extremely slow. Is it possible to adopt this model posterior to further speed up the encoding of samples out of the training set? i.e. we may imagine efficiently finetuning a new INR (posterior) from the obtained prior. If this finetuning requires less time to convergence, the enlarged encoding time may be amortized.

Another issue for me is the somewhat marginal performance improvement. Is it worth costing such a large encoding time for the model prior, instead of simply training a COIN/COIN++ model longer to compensate for the quantization error? It would be better to report the converging speed of the models e.g. something similar to PSNR-iteration curves.

**Questions:**

See above

**Limitations:**

The authors have discussed the limitations, which is convincing.

---

> ### Author Rebuttal · Authors · 2023-08-09
>
> We thank the reviewer for the thoughtful comments and feedback on our work; we address the reviewer's concerns below.
>
> > Seems that to train a model prior, we should first train many model posteriors. As also discussed by the authors, this training makes the entire training (encoding) time extremely slow. Is it possible to adopt the model posterior of training samples to further speed up the encoding of samples out of the training set? We may imagine efficiently finetuning a new INR (posterior) from the obtained prior.
>
> This appears to be a misunderstanding; we don't need to re-learn the prior each time we encode an image. We learn the model prior once from a few training images and their corresponding posteriors, akin to learning the weights of a neural network before deployment. Then, during test time, we fix the prior we learnt earlier and only optimize the variational posterior of the INR corresponding to the data.
>
> We experimented with training a new INR model initialized from the established prior. However, we observed its convergence speed is similar to an appropriately randomized initialization. Consequently, given a model prior and a test image, we trained the corresponding INR posterior from scratch. However, it is an interesting question whether we could find better initialization heuristics for our INRs to speed up convergence. We will note this as a possible avenue for future work in the camera-ready version of the paper.
>
> > Is it worth costing such a large encoding time to compensate for the quantization error?
>
> With the sacrifice of encoding time, we contribute to two critical improvements compared with previous methods: (1) joint rate-distortion optimization; (2) memory-efficient training. The first improvement is critical because the development path of VAE-based compression indicates that joint rate-distortion is necessary for superior performance. The second improvement means we do not need to crop high-resolution signals such as Kodak images into patches like previous methods. In that way, implicit neural representations have much more potential to capture the whole image's correlation structure and deliver better compression performance.
>
> Compression with implicit neural representations is a new and promising paradigm in data compression. Many researchers are currently interested in whether INR-based compression can surpass VAE-based compression in terms of rate-distortion performance. It is quite similar when Minnen et al. [3], for the first time, employed the autoregressive context model to improve the compression performance, which increases the decoding time dramatically. But several follow-up works successfully solved this issue soon after the work of Minnen et al. [3]. We believe our work can provide insights into this field as a new INR-based compression approach with several advantages, and we are hopeful that many of our method's practical limitations will be removed in future work.
>
> > It would be better to report the converging speed of the models e.g. something similar to PSNR-iteration curves.
>
> We appreciate your constructive suggestion. We conducted additional experiments to address this, adjusting the finetuning strategy across varying iteration numbers. Our analysis on a Kodak image (Kodim03.png) is described in a graph juxtaposing PSNR against iteration number. Figure 1 in our rebuttal document shows that performance improves marginally as iterations increase, and the finetuning time grows linearly according to the iteration number. The results in Figure 2b of our main paper match the highest iteration number. Notably, we can reduce encoding time at the small cost of a 0.3dB drop in PSNR by setting the iteration number to 10%.

---

> > ### Comment · Reviewer_7dCG · 2023-08-21
> >
> > The authors have addressed my concerns. I maintain my acceptance rating.

---

### Official Review · Reviewer_D8f2 · 2023-07-15

**Soundness:** 4 excellent
**Presentation:** 4 excellent
**Contribution:** 4 excellent
**Rating:** 8
**Confidence:** 3

**Summary:**

This paper addresses the problem of lossy data compression (evaluation is on images and audio) using implicit neural representations (INRs). In this approach, a neural network is designed that maps coordinates (e.g., x,y locations in an image or time for audio) to samples (RGB values or audio amplitude). Then the overfit network itself is the representation that is stored or transmitted since the data can be approximately recovered by running the network over appropriate coordinates.

INRs for data compression are not new, but this paper includes two key novelties: (1) a variational Bayesian approach is adopted, which (2) allows the method to jointly optimize for rate (the entropy of the neural network parameters) and distortion (the quality of the reconstructed data). Previous INRs for compression used less powerful formulations and separately optimized for rate and distortion, which typically leads to worse overall performance (and the empirical evaluation in this paper demonstrates that here).

Within this setup, the authors describe an iterative algorithm for learning the prior over network weights, and they describe a progressive refinement method that improves performance by dividing the network parameters into blocks and conditionally coding each block given previous blocks.

The method is evaluated on images (CIFAR-10 and Kodak) and audio (LibriSpeech). In both cases, the method is shown to outperform previous INR-based models. For images, it does not outperform the best VAE-based compression methods.

**Strengths:**

Originality: the problem is not new and jointly optimizing rate and distortion is quite common in neural compression in general, but I have not seen it done before for INRs. Nor have I seen an approach that uses variational Bayes and relative entropy coding with INRs.

Quality and clarity are both excellent.

I think the significance of this paper is quite high for the neural compression community and especially researchers looking at INRs. VAE-based (often called "nonlinear transform coding" or NTC) is the dominant approach right now, but INRs are quite interesting and there's a growing literature using such methods for compression. This work is significant because it does the right thing (joint optimization of rate and distortion), which has not been done before with INRs, and because it sets a new SOTA for compression with this approach.

**Weaknesses:**

1. Obviously the paper would be stronger if the empirical results were better. Focusing on Fig. 2b (RD curves on the Kodak image set), COMBINER trails CVPR2020 by more than 2dB at 0.2 bpp (a huge gap), and CVPR2020 is no longer a SOTA approach. That said, COMBINER is SOTA for INR-based methods (as far as I know), which is an important result, and I think research can get stuck in a local minimum if we reviewers require new approaches to outperform established ones too soon in the research cycle, e.g., Fig. 2b shows that COMBINER slightly outperforms the VAE-based method from ICLR2018.

2. Neural model compression is a large subfield so some discussion or comparison with a model compression method (beyond simple weight quantization) seems like an important baseline that's missing.

**Questions:**

Section 5.1 ("Models") discusses the model architecture saying that it was empirically determined. How sensitive are the results to the base model architecture, and how easily can the encoding process prune (reduce parameter entropy to near zero) large parts of a model architecture that is much larger than the RD optimal model?

**Limitations:**

adequately addressed

---

> ### Author Rebuttal · Authors · 2023-08-09
>
> We thank the reviewer for the careful reading and constructive feedback and respond to the reviewer's concerns below.
>
> > Obviously, the paper would be stronger if the empirical results were better. COMBINER is SOTA for INR-based methods (as far as I know), which is an important result, and I think research can get stuck in a local minimum if we reviewers require new approaches to outperform established ones too soon in the research cycle.
>
> Thanks for the positive comments. As the reviewer says, it is often unrealistic to expect a new approach to surpass all these well-established approaches in a short time. But we do believe that our paper will provide insights into this field, and we will keep improving our method continually. We believe this framework will soon be developed to achieve more impressive compression performance.
>
> > Neural model compression is a large subfield so some discussion or comparison with a model compression method (beyond simple weight quantization) seems like an important baseline that's missing.
>
> Havasi et al. [23] applied relative entropy coding to model compression and demonstrated state-of-the-art performance, and they also compared it with some model compression baselines. Another paper [Ref1] also compared with some model compression methods and found using relative entropy coding is more effective for Bayesian model compression than other model compression methods, especially when the model size is small. This is shown, for example, in Table 1 of [Ref1] (Minimal Random Coding Learning achieves the best performance on LeNet5-Caffe). As these prior works already demonstrate the superior performance of using relative entropy coding for model compression, we omit these results in our paper.
>
> ## References
>
> [Ref1] Scalable Model Compression by Entropy Penalized Reparameterization. Oktay et al., ICLR 2020.

---

> > ### Comment · Reviewer_D8f2 · 2023-08-22
> >
> > > Havasi et al. [23] applied relative entropy coding to model compression and demonstrated state-of-the-art performance...
> >
> > The authors' reliance on previous results comparing relative entropy coding to other model compression methods seems sufficient. It would be good to add a reference to (Oktay 2020) and make it clear in the paper that they're relying on results from that paper and (Havasi 2019).
> >
> > I don't see any reason to adjust my rating.

---

### Official Review · Reviewer_WLVL · 2023-07-27

**Soundness:** 3 good
**Presentation:** 3 good
**Contribution:** 2 fair
**Rating:** 5
**Confidence:** 2

**Summary:**

The paper proposes a new method for compressing general signals, by using Variational Bayesian implicit neural representations. It proposes an algorithm for learning a prior distribution over the implicit representation weights, as well as a pipeline for inferring the posterior distribution corresponding to every given signal to be encoded. The authors employ several tricks (e.g. Relative Entropy Coding) borrowed from previous works, that are meant to increase compression efficiency.

The method is evaluated over two image datasets (namely the CIFAR10 dataset of very small images and the Kodak dataset containing 24 larger images) and one audio dataset, while comparing performance with several image compression methods and the classic MP3 audio compression method, respectively.

**Strengths:**

Generally speaking, the paper is presented well - despite the fact that compression is not my main expertise, I was able to follow and understand most of it.
The combination of the different ideas used in this method is interesting and appealing.
Limitations, as well as concurrent work, are discussed candidly.

**Weaknesses:**

Besides the method's complexity, which could (at least partially) be attributed to my lack of expertise in this field, I found two major flows:

Performance:
The performance curves presented in Fig. 2 and 5 in the paper do not indicate an advantage of the proposed method over many of the existing methods, in terms of the rate-distortion trade-off. In the case of audio (Fig. 5), some advantage is shown only over the classic MP3 compression, and only for part of the kbps range.

Speed/Complexity:
The encoding process of this method is long and expensive. Besides being a big limitation (as noted by the authors) and hence a weakness of the method compared to some of the other ones, I'm missing some detailed comparison of these aspects.

A third, relatively more minor flow has to do with the experimental setup itself, as experiments were conducted over only two small image datasets and one audio dataset, and compared only with MP3 in the latter case.

**Questions:**

As I'm not entirely familiar with this field, the authors are welcome to direct my attention to certain points they feel that I overlooked, and I'll gladly consider increasing my initial rating.

**Limitations:**

Limitations were candidly discussed in the paper.

---

> ### Author Rebuttal · Authors · 2023-08-09
>
> We appreciate the reviewer's time and effort to review our paper and address your concerns below.
>
> ## Performance
>
> In our paper, we focus on beating the previous INR-based compression methods. The performance of VAE-based methods is shown in Figure 2 using dotted lines for reference, similar to all previous INR-based compression papers [10][11][12].
> While VAE-based methods have long been dominant in neural image compression, compressing with INRs has emerged as a promising alternative but has certain limitations given its novelty and short development history. As Reviewer D8f2 commented, research would get stuck in a local minimum if we require new approaches to outperform established ones too soon in the research cycle. Our proposed method supports joint rate-distortion optimization, a significant improvement compared to previous INR-based compression methods.
>
> ## Complexity
>
> First, after improving the practical implementation of entropy coding, we update our method's encoding and decoding times in Tables 1 and 2, as shown in the one-page rebuttal document. Now both the encoding time and decoding time are acceptable for practical application. Specifically, on the CIFAR-10 dataset, it only takes around 33 minutes for our largest bitrate model to encode 500 images from the CIFAR-10 dataset, meaning each CIFAR-10 image takes around 3.96 seconds for encoding. We can compare our method COMBINER with previous methods like COIN [10] and COIN++ [11] (according to the coding time reported in the paper of COIN++), showing the comparisons in the below table.
>
> |                   |  COIN             |  COIN++           |  COMBINER (Ours)  |
> |-------------------|-------------------|-------------------|-------------------|
> | Encoding Time     |    2.97 s         |    0.095 s        |    3.96 s         |
> | Decoding Time     |    0.46 ms        |    1.29 ms        |    3.90 ms        |
>
> Second, we can reduce the number of iterations during the finetuning process to shorten the encoding time. As shown in Figure 1 in the rebuttal document, we can reduce the iteration number from about 30000 to less than 3000, while the PSNR only drops for 0.3 dB. It demonstrates the time of progressive finetuning, which constitutes the bulk of the encoding time, can be largely reduced by only sacrificing a little compression performance.
>
> ## Experimental Setup
>
> Evaluating on Kodak dataset with 24 images is a standard setting for research in data compression. It is because compression models are not so sensitive to imbalanced data distribution, and the image distribution on the Kodak dataset is relatively representative.
> In addition, most previous INR-based works are constrained by MAML's computational demands and hence cannot be directly applied to high-resolution image compression like COMBINER can.
> Therefore, these works report their performance on the low-resolution Cifar-10 dataset.
> Since these works are our important baselines, we also follow this setting and compare the results on the Cifar-10 dataset for consistency.
>
> _We are happy to answer any further questions the reviewer might have. However, if we answered the reviewer's questions adequately, we kindly invite the reviewer to consider raising their score._

---

> > ### Comment · Reviewer_WLVL · 2023-08-17
> >
> > Thank you for your response.
> > I now feel comfortable raising my score, following your clarification regarding the distinction with regard to INR-based methods (due to their relative novelty). However, as this argument still does not explain the slight disadvantage with respect to the INR-based MSCN method on the CIFAR dataset, and since it's unclear whether the proposed method is better than MSCN in terms of speed (as noted by reviewer rYFn), I'll settle for borderline accept.

---

### Author Rebuttal · Authors · 2023-08-09

We extend our gratitude to all the reviewers for their comprehensive feedback and time spent reviewing our manuscript. It is heartening that all the reviewers agree that the idea proposed in this paper is novel and valuable. We have addressed their concerns in our respective responses.

In addition, after submitting the manuscript, we cleaned up the code and improved the practical implementation. On the one hand, we attach an anonymous link of our code to the AC for reproducibility guided by the NeurIPS rebuttal policies. On the other, we also update the practical encoding and decoding time, as shown in Tables 1 and 2 of the one-page rebuttal document. Specifically, encoding for 500 CIFAR-10 images now ranges from 13 to 33 minutes (0.91 to 4.45 bpp), which means it only takes a few seconds to encode each CIFAR-10 image. For high-resolution Kodak images, the encoding time is now only 21.5 minutes at 0.07 bpp. Moreover, by adjusting the number of iterations during the progressive finetuning phase, we can reduce the finetuning time by 90 %, with only a minimal 0.3 dB performance trade-off. We will include these statistics in the final version of our paper.

---

### Decision · Program_Chairs · 2023-09-21

**Decision:**

Accept (spotlight)

**Comment:**

Author rebuttal addressed most of the reviewer concerns and all reviewers are leaning accept after final discussion.

While implicit representations are catching on for compression applications, the paper proposes to fit a variational BNN and then uses relative entropy coding (as opposed to quantization and entropy coding, which is most common). This combination is quite unique and flexible, which in conjunction with other contributions, yields strong results worthy of highlighting at the conference via spotlight.